# Extreme genetic signatures of local adaptation during *Lotus japonicus* colonization of Japan

Niraj Shah [1,12], Tomomi Wakabayashi [2,12], Yasuko Kawamura [3,12], Cathrine Kiel Skovbjerg [1], Ming-Zhuo Wang[3], Yusdar Mustamin[3], Yoshiko Isomura[3], Vikas Gupta[1], Haojie Jin [1], Terry Mun [1], Niels Sandal[1], Fuyuki Azuma[4], Eigo Fukai [4], Ümit Seren [5], Shohei Kusakabe[3], Yuki Kikuchi[3], Shogo Nitanda[3], Takashi Kumaki[3], Masatsugu Hashiguchi[6], Hidenori Tanaka [6], Atsushi Hayashi[7], Mads Sønderkær [8], Kaare Lehmann Nielsen [8], Korbinian Schneeberger [9], Bjarni Vilhjalmsson [10], Ryo Akashi[6], Jens Stougaard [1], Shusei Sato[3]*, Mikkel Heide Schierup [11]* & Stig Uggerhøj Andersen [1]*

Colonization of new habitats is expected to require genetic adaptations to overcome environmental challenges. Here, we use full genome re-sequencing and extensive common garden experiments to investigate demographic and selective processes associated with colonization of Japan by *Lotus japonicus* over the past ~20,000 years. Based on patterns of genomic variation, we infer the details of the colonization process where *L. japonicus* gradually spread from subtropical conditions to much colder climates in northern Japan. We identify genomic regions with extreme genetic differentiation between northern and southern subpopulations and perform population structure-corrected association mapping of phenotypic traits measured in a common garden. Comparing the results of these analyses, we find that signatures of extreme subpopulation differentiation overlap strongly with phenotype association signals for overwintering and flowering time traits. Our results provide evidence that these traits were direct targets of selection during colonization and point to associated candidate genes.

[1] Department of Molecular Biology and Genetics, Aarhus University, 8000 Aarhus C, Denmark. [2] Graduate School of Human and Environmental Studies, Kyoto University, Yoshida-nihonmatsucho, Sakyo-ku, Kyoto 606-8501, Japan. [3] Graduate School of Life Sciences, Tohoku University, Katahira 2-1-1, Aoba-ku, Sendai 980-8577, Japan. [4] Laboratory of Plant Breeding, Graduate School of Science and Technology, Niigata University, 2-8050 Ikarashi, Nishi-ku, Niigata 950-2181, Japan. [5] Gregor Mendel Institute, Austrian Academy of Sciences, Vienna B(VBC), 1030 Vienna, Austria. [6] Faculty of Agriculture, University of Miyazaki, Gakuen-Kibanadainishi 1-1, Miyazaki 889-2192, Japan. [7] Kazusa DNA Research Institute, Kazusa-Kamatari 2-6-7, Kisarazu 292-0818, Japan. [8] Department of Chemistry and Bioscience, Section for Biotechnology, Aalborg University, 9220 Aalborg, Denmark. [9] Max Planck Institute for Plant Breeding Research, 50829 Cologne, Germany. [10] National Centre for Register-based Research, Department of Economics and Business Economics, Aarhus University, 8210 Aarhus V, Denmark. [11] Bioinformatics Research Centre, Aarhus University, 8000 Aarhus C, Denmark. [12] These authors contributed equally: Niraj Shah, Tomomi Wakabayashi, Yasuko Kawamura. *email: shuseis@ige.tohoku.ac.jp; mheide@birc.au.dk; sua@mbg.au.dk

When populations evolve adaptations that are advantageous in their local environment, irrespective of the effects of these adaptations in other environments, they become locally adapted. Experimental evidence for local adaptation can be obtained by quantifying the fitness of individuals from two or more populations in common garden experiments at two or more locations. If population A outperforms population B when grown in a common garden in population A's native environment, and the converse is true in population B's native environment, it indicates that populations A and B are locally adapted to their respective native environments[1]. Such experiments are now being coupled with genotype data to begin understanding the genetics of local adaptation[2]. Once it has been demonstrated that a population displays local adaptation, genotyping of individuals from contrasting populations allows genome scans for selection signatures such as high fixation index ($F_{ST}$) levels. Demographic history, however, can generate similar signatures through genetic drift, causing false positives.

Population structure, whether due to genetic drift or selection, has long been recognized as a major confounding effect in genome-wide association (GWA) studies, and stringent population structure correction methods have been developed to separate genuine genotype-phenotype associations from spurious associations due to population structure[3,4]. It has been suggested that combining population structure-corrected GWA analysis of phenotypic data from common garden experiments with purely genotype-based genome scans could be a powerful approach for studying local adaptation. If a potentially adaptive signal detected using a phenotype-independent genome scan for population differentiation overlaps with a GWA signal derived from common garden fitness data, the argument is that these signals would constitute two independent lines of evidence supporting local adaptation, one based on genomic and the other on phenotypic differentiation[5,6].

Perhaps the most striking example of GWA application to the study of local adaptation is the investigation of human skin pigmentation. Evolution of skin pigmentation is driven by UV irradiation, with dark skin offering protection where there are high levels of UV irradiation, and light skin promoting vitamin D synthesis under low UV irradiation. SNPs in several genes involved in melanin production were identified as associated with skin pigmentation in GWA scans, and they also generally show extremely high $F_{ST}$ levels[7]. Relatively few studies have applied GWA to the investigation of plant local adaptation. One large Arabidopsis study used common garden fitness data in combination with GWA and correlations with climatic factors, but did not employ genome scans for population differentiation[2]. A second study examined drought stress data from a laboratory experiment and observed marginally elevated $F_{ST}$ levels for the top SNPs[8], while a third showed evidence for an adaptive role of a sodium transporter by GWA analysis of ion accumulation coupled with investigation of gene expression and distances to saline soils[9]. In addition, flowering time is considered a critical adaptive trait in annual Arabidopsis and has been thoroughly examined in GWA studies[3,10]. However, human dispersal of Arabidopsis seeds has weakened the link between geographic origin and plant genotype, complicating interpretation with respect to local adaptation[10–12]. Most of the relatively few potentially adaptive plant traits studied by GWA have thus not revealed clear signals of local adaptation, and the power of combining GWA analysis of common garden phenotype data with genotype-based genome scans for the study of local adaptation has not been systematically explored.

Lotus japonicus (Lotus) is a self-compatible, diploid, wild perennial legume[13]. It has a relatively small genome size of ~472 Mb and is found in diverse natural habitats across East and Central Asia, including Japan, Korea, and China, and extending west into Afghanistan[14]. L. japonicus natural diversity has been examined through the establishment of a collection of wild Japanese accessions[14]. They represent an interesting population sample for studying local adaptation because of the geographical isolation of the Japanese archipelago and the pronounced variation in climate between the southern and the northern parts of the country. Climates range from subtropical to hemiboreal with yearly average temperatures of 17.9 and 5.3 °C, respectively, while annual daylight varies from 1316 to 2202 h, and annual precipitation ranges from 775 mm to 3250 mm[13]. The topography of Japan is also very varied, and an up to 3000 meters high mountain range traverses the central regions of the archipelago. The mountainous topography will likely limit dispersal and the steep climatic gradients could result in strong selection pressures, creating conditions that could promote population differentiation driven by local adaptation[1].

Here, we genetically characterize a set of wild Japanese Lotus accessions, identify subpopulations, characterize their demographic history, and show evidence for local adaptation to a cold climate in common garden experiments. We discover very pronounced overlaps in adaptive signals between phenotype-based GWA and genotype-based $F_{ST}$ approaches, allowing us to identify traits and associated genomic loci that were likely direct targets of selection during local adaptation.

## Results

**Kyushu Island is the center of Lotus diversity in Japan.** We carried out whole-genome re-sequencing of 136 wild Lotus accessions collected throughout Japan using Illumina paired-end reads (Fig. 1a, Supplementary Data 1), identifying a set of 525,800 high confidence SNPs in non-repetitive regions (Supplementary Figs. 1, 2). In order to examine the genetic relatedness of the accessions, we carried out a principal component analysis (PCA) based on the accession genotypes. We found a striking correlation between the position of the accessions in the PCA plot and their geographical origin (Fig. 1a, b), similar to that observed for studies of human populations[15,16]. PC1 clearly separated the north and south of Japan, while PC2 most strongly distinguished accessions from the eastern and southern coastline of Kyushu Island (Fig. 1a, b). For both PC1 and PC2, the northern accessions clustered more tightly than the southern accessions, and the accessions from Kyushu Island were particularly well-resolved (Fig. 1a, b).

The tight clustering of the northern accessions suggested that these could have lower levels of genetic diversity and, consequently, that Lotus could have arrived in the south of Japan and then migrated north. To examine the migration history of Lotus in detail, we carried out a Pairwise Sequentially Markovian Coalescent (PSMC) model analysis, which infers population size history from a diploid sequence based on the extent of heterozygous regions in the genome[17]. Lotus is self-compatible and based on the accession heterozygosity levels (Supplementary Data 1), we estimate that is has a selfing rate of 94%. The individual accessions were therefore not well suited for PSMC analysis. Instead, we generated pseudo-diploids by merging read alignments from pairs of a subset of individuals with at least 7x read coverage and called a pseudo-diploid consensus sequence based on the merged alignments, which was used for PSMC analysis. This type of analysis can infer the time of last contact of the pair of individuals forming the pseudo-diploid as the time of the last coalescence events, measured as an abrupt increase in the estimated effective population size. We exploited this feature to roughly estimate divergence times (cessation of gene flow) for the pseudodiploid pairs as the point in time when the PSMC curve

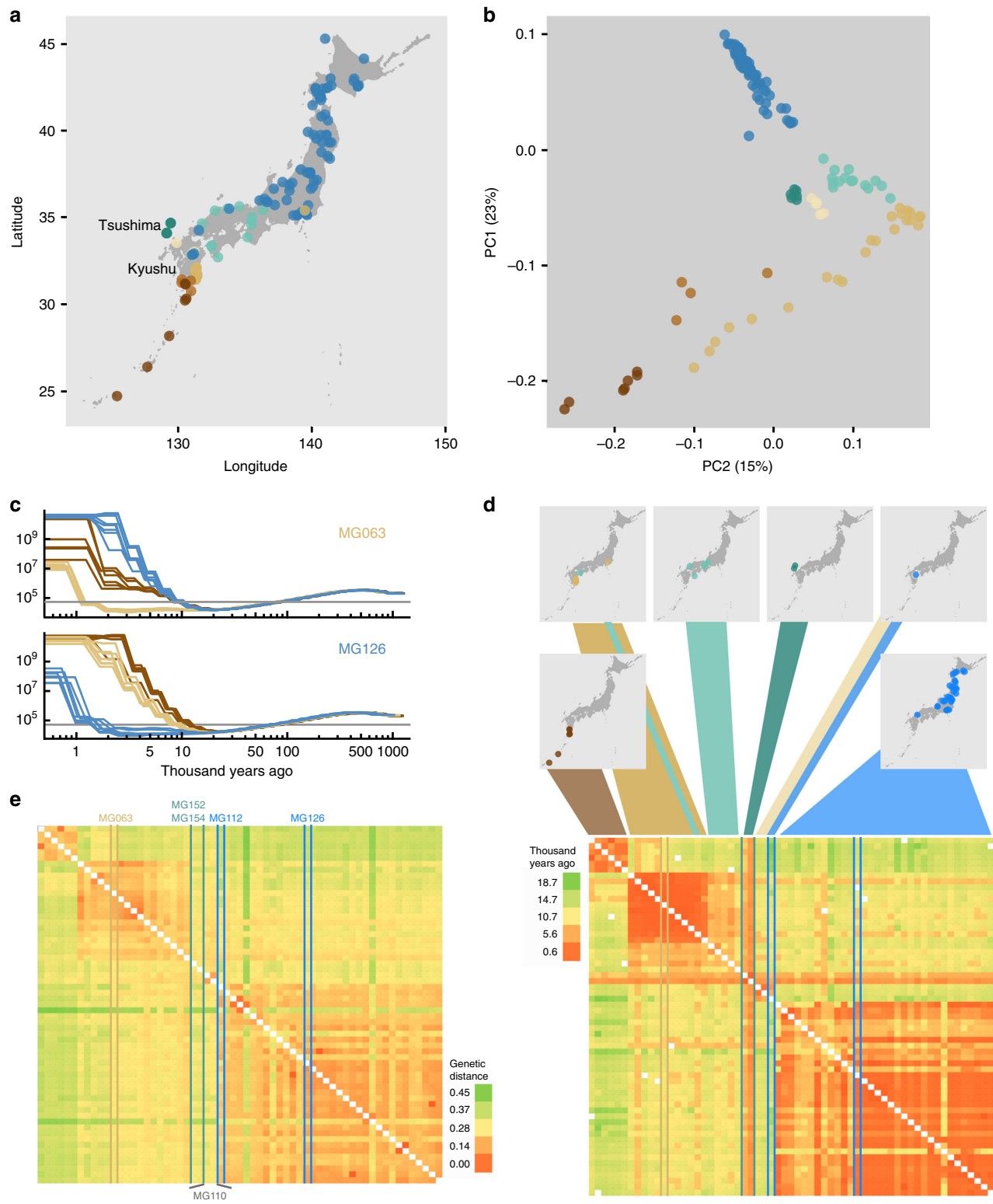

**Fig. 1 Geography and genetics of Japanese *Lotus* accessions. a** Geographical origin of the *Lotus* accessions. **b** Principal component analysis (PCA) based on the genotypes of the *Lotus* accessions. **a–b** Accessions are colored according to their grouping in the PCA plot. **c** Pairwise sequentially Markovian coalescent (PSMC) curves based on pseudo-diploids generated by merging alignments of pairs of accessions. Each curve indicates the inferred population size history through time for pseudodiploid pairs including MG063 and MG126, respectively. The horizontal gray line indicates an effective population size of $5 \times 10^4$. **d** The lower panel shows a heatmap of divergence times (years ago) for pseudodiploid pairs as estimated from the PSMC analysis with a population size cutoff of $5 \times 10^4$. Accession colors are matched in **a–e**. **e** Heatmap showing the genetic distances between pairs of accessions. Genetic distance indicates the average number of allelic differences per polymorphic site in a pairwise comparison. **d, e** Accessions MG152 and MG154 from Tsushima Island (green), MG110 and MG111 (gray) and MG112 (blue) from central Kyushu, MG126 from northern Japan, and M063 from eastern Kyushu are highlighted as indicated in **e**. The accessions are shown in the same order in **d** and **e**. Source data underlying **d** and **e** are provided as a Source Data file.

abruptly rises (Fig. 1c), assuming a mutation rate of $6.5 \times 10^{-9}$ per year[18]. For example, for accession MG126 found in northern Japan, the cessation of gene flow to other accessions from northern Japan (blue lines) are 1–5 thousand years, for Eastern Kyushu (light brown lines) 7–9 thousand years and for south and west Kyushu (dark brown lines) 10–12 thousand years (Fig. 1c). The estimated divergence times for all pairs of accessions showed clear clustering along the inferred colonization route (Fig. 1d) with South and Northern Japan having last contact 10–18 thousand years ago, consistent with colonization after the last ice age. Note that although the PSMC estimation is inaccurate for the past 10,000 years when applied to data from humans with a generation time of roughly 30 years, it should be accurate up to about 10,000/15 to 30 = 333 to 667 years ago for *Lotus*, assuming a *Lotus* generation time of one to two years. Note that the generation time does not affect the time estimates for cessation of gene flow, since these are based on the mutation rate per year and mutations accumulate continuously throughout the lifetime of the plant.

For the same accession pairs used in the PSMC analysis, we also calculated simple genetic distances (Fig. 1e). Despite the clear overall similarity of the genetic distance and PSMC-based divergence time analysis, we found marked deviations for accessions from Tsushima Island (MG152 and MG154) and central Kyushu (MG112). The Tsushima accessions appeared to have diverged recently from all other accessions (Fig. 1d), but the genetic distance analysis indicated relatively large genetic distances from the Tsushima lines to all other accessions with no strong association to any other population cluster (Fig. 1e). This unique signature and the central location of these accessions in the PCA plot (Fig. 1b), suggests that Tsushima Island may represent a point of origin for all of the Japanese *Lotus* populations. Among the central Kyushu lines, MG112 is the southernmost accession belonging to blue cluster in the PCA plot (Fig. 1a, d). In the genetic distance analysis, MG112 is clearly most similar to the other accessions from this PCA cluster, but it shows long divergence times from these and shorter divergence times from its fellow central Kyushu lines (Fig. 1d, e). Likewise, MG110 and MG111 show long divergence times from other accessions in their geographical vicinity (except for the Tsushima lines), but shorter divergence times with the central Kyushu lines (Fig. 1d, e). These central Kyushu lines, which grow at high altitudes with sub-0 °C minimum temperatures, thus likely became isolated and stopped exchanging genetic material with other accessions relatively shortly after *Lotus* colonization of Japan. Altogether, our data suggests that *Lotus* migrated north and south from a starting point near Tsushima Island and that the north was colonized most recently, since accession pairs across a large geographical area in northern Japan display relatively short divergence times (Fig. 1d). Our results also indicate that representatives of all major genetic clusters are found on Kyushu Island, making it the center of diversity for *Lotus* in Japan.

### Three *Lotus* populations exist in Japan

In the PCA analysis, the northern and southern accessions clustered into different groups, indicating that distinct subpopulations might exist. fastSTRUCTURE[19] analysis, which groups accessions based on allele frequencies, suggested that a large fraction of the variation could be accounted for by three subpopulations (Supplementary Fig. 3). Populations 1 (pop1), 2 (pop2), and 3 (pop3) occupied southern, central, and northern Japan, respectively, and corresponded to the three vertices in the PCA plot (Fig. 2a–c). The three subpopulations also overlapped with the major clusters identified in the analysis of pairwise genetic distances and

divergence times (Fig. 1d, e), indicating that they reflect a robust grouping of the accessions. This is supported by Fig. 2d, which shows that the genetic distance for a given physical distance is larger for comparisons between these three populations as compared to distances within populations. Examining genetic diversity (pi) for each subpopulation, we found that the southernmost populations showed higher levels of diversity (Fig. 2e), which is consistent with the hypothesis that *Lotus* first migrated south from a starting point near Tsushima Island and then later colonized the north of Japan, losing genetic diversity along the way.

### The populations show extreme genetic differentiation

We calculated $F_{ST}$ for all polymorphic positions, comparing pop3 accessions against accessions with no population 3 membership in order to detect markers strongly differentiated between these two groups with relatively many characterized members. There were 5,612 genes with at least four informative SNPs, for which we calculated the mean $F_{ST}$ per gene. The median per-gene $F_{ST}$ value was 0.24, and the top gene *Lj6g3v1790920* spanned 1.2 kb and had a mean $F_{ST}$ value of 0.99 across 20 SNPs (Supplementary Fig. 4). It was located within a ~20 kb genomic region showing very strong fixation of the alternative allele in pop3 relative to accessions without pop3 membership (Supplementary Fig. 4C).

### Population 3 is adapted to a cold climate

The distinct regions occupied by each of the three subpopulations, the restriction of pop3 accessions in southern Japan to a cold, high-altitude environment in central Kyushu, and the pronounced differentiation observed for specific genomic regions, suggested that population differentiation could have been influenced by local adaptation in addition to genetic drift. To test this hypothesis experimentally, we grew the accessions in a common garden using a field site at Tohoku University (38.46 °N, 141.09 °E), which is located in northern Japan where pop3 individuals dominate (Figs. 2a and 3a). We saw pronounced variation in flowering time, and a number of plants died during winter (Fig. 3). We grew the accessions for four consecutive years at the field site, and each year we replanted to replace the plants that did not survive winter. When quantifying phenotypic traits, only plants of the same age were compared and we only scored plants that were well established in fall for winter survival (Fig. 3a).

To test for evidence of local adaptation of pop3 to its native northern Japan environment, we analyzed winter survival, which provides a direct fitness measure, as a function of population membership for non-admixed accessions. In the field, the accessions were planted in blocks of three in a row and column grid, resulting in a data structure with block nested within accession, which again is nested within population (Population: Accession:Block). Field row and column placement each accounted for only ~1% of the variation in survival phenotype, the Population:Accession and Population:Accession:Block terms each explained ~8%, while Population explained 32% of the variation. From 2014 to 2016, pop3 accessions clearly outperformed both pop1 and pop2 individuals in terms of survival ($p < 1e-4$, mixed linear model ANOVA with Tukey multiple comparison test) (Fig. 4a). In 2017 an earlier planting date and milder winter resulted in survival of nearly all accessions, regardless of population membership (Fig. 4a).

The better survival of pop3 relative to the other populations suggested that it could be locally adapted to a cold climate, but it was also possible that pop3 was simply more hardy in transplantation experiments in general. To investigate if this was the case, we grew eight *Lotus* accessions, including two pop3 accessions, in a field at University of Miyazaki (31.83°N, 131.41° E) (Fig. 4b). The northernmost (MG030 and MG007) and

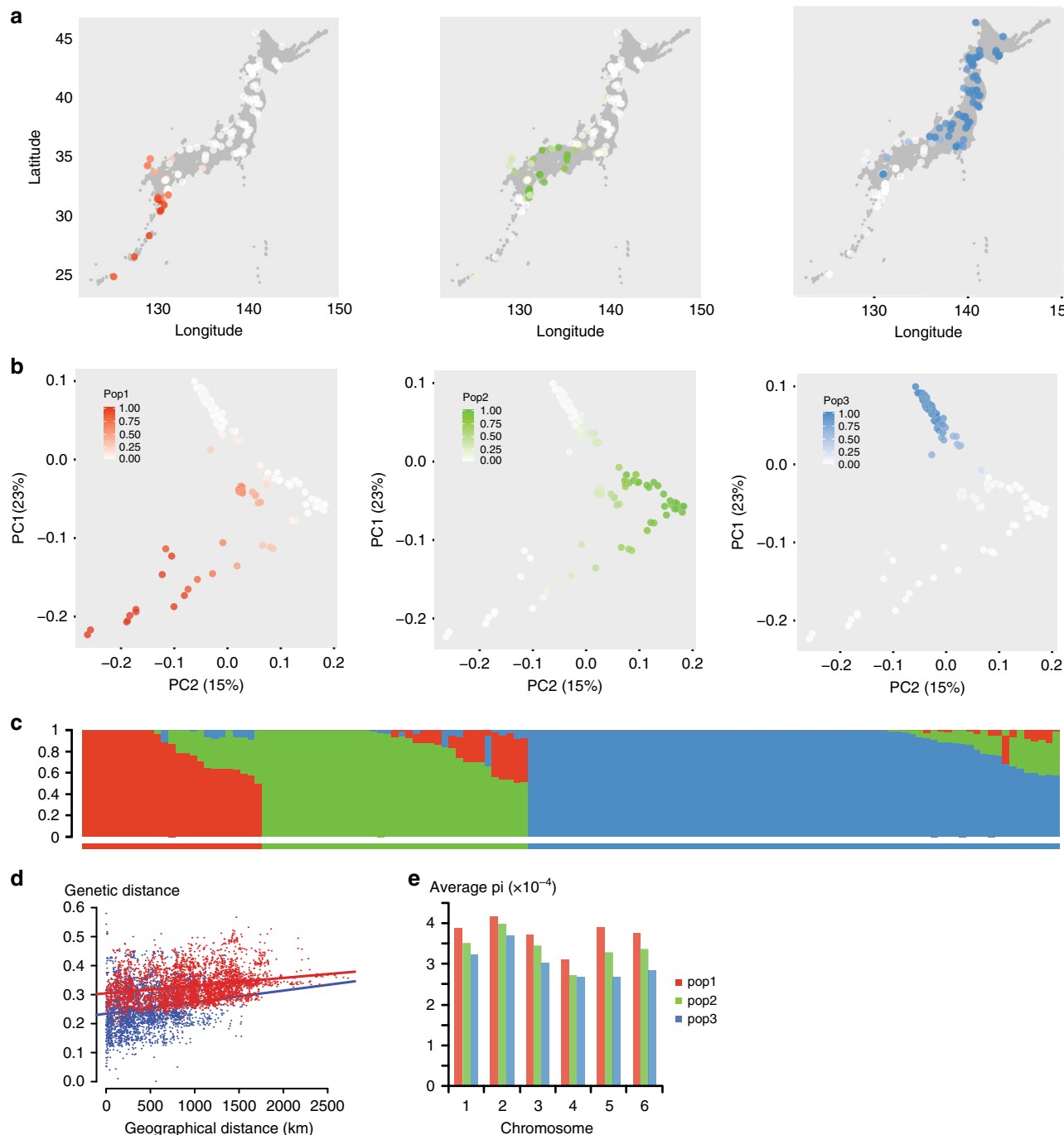

**Fig. 2 Population structure and genetic diversity. a–b** Accessions are colored by their population membership placed according to geographical origin (**a**) or according to their coordinates in the PCA analysis (**b**). **c** STRUCTURE plot showing population memberships for pop1 (red), pop2 (green) and pop3 (blue). **d** Genetic distance versus geographical distance. Genetic distance indicates the average number of allelic differences per polymorphic site in a pairwise comparison. Red: between-population comparisons. Blue: within population comparisons. **e** Average nucleotide diversity (pi) per chromosome.

southernmost (MG020 and MG066) accessions displayed markedly different survival rates at Miyazaki and Tohoku, while the accessions from central Japan showed less pronounced differences in performance at the two locations (Supplementary Table 1). To investigate population level effects, we compared pop3 individuals (MG030 and MG007) to those with no pop3 membership (MG008, MG066, and MG020) (Supplementary Table 1). These two groups of accessions showed Tohoku survival rates comparable to the means of all pop3 and non-pop3 accessions, respectively (Supplementary Tables 2–5). The pop3 accessions performed better than non-pop3 individuals at the

northern Tohoku site, while the converse was true at the southern Miyazaki site (Fig. 4c). There was a highly significant location by population interaction ($p = 3 \times 10^{-9}$, mixed linear model ANOVA) and differences were significant for pop3 vs. non-pop3 survival at Tohoku ($p = 0.041$) and Miyazaki ($p < 0.020$, mixed linear model ANOVA with Tukey multiple comparison test) (Supplementary Tables 2–5). The pop3 individuals grown at Miyazaki appeared stressed during summer, where they dropped a number of leaves and showed significant damage already in August (Fig. 4d). This is in contrast to the southern pop1 and pop2 individuals grown at Tohoku, which were generally healthy

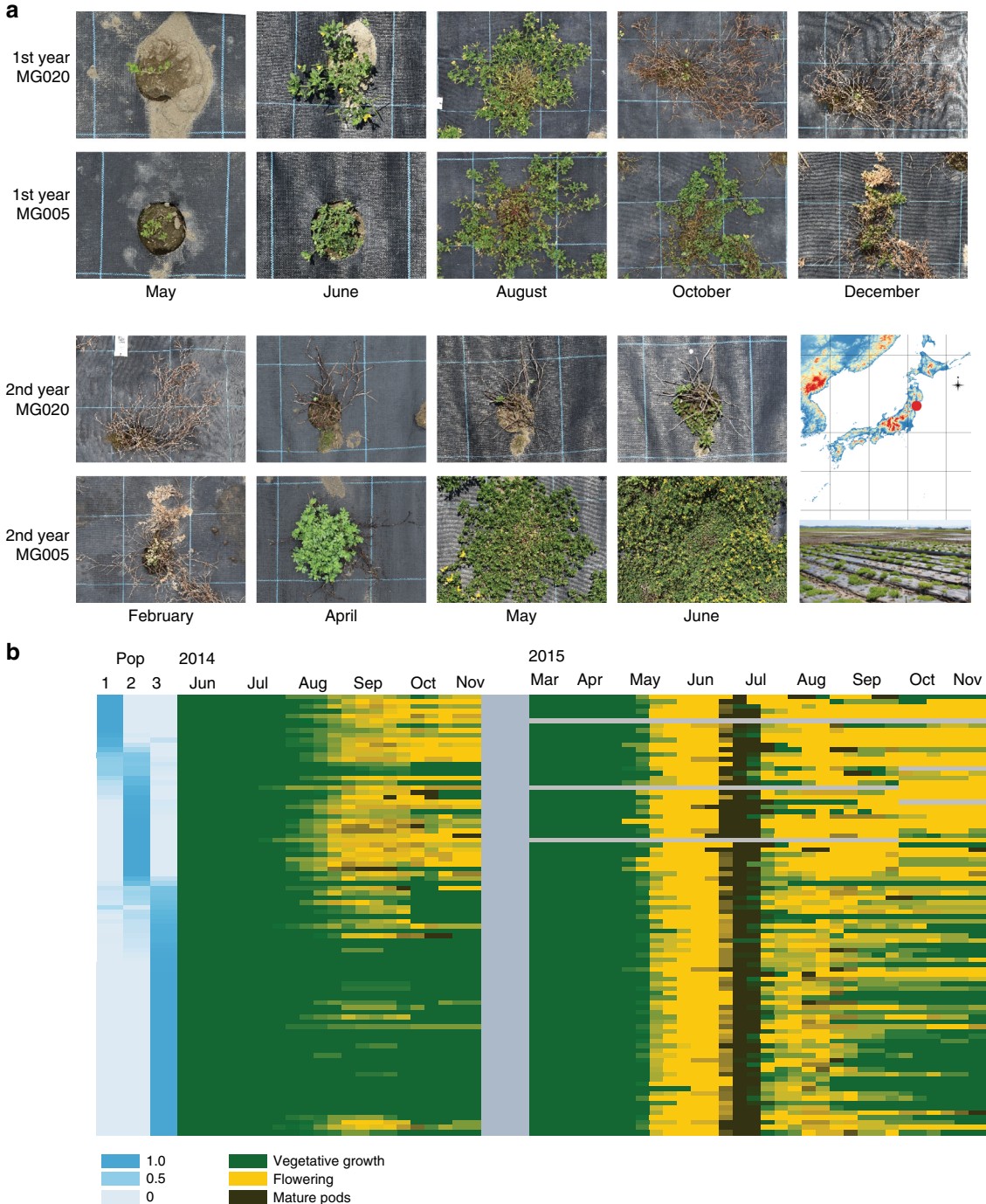

**Fig. 3 Overwintering and flowering time phenotypes. a** Pictures from the field site at Tohoku, 38.46 N, 141.09 E, 2017/2018. MG020 is from Miyako Island in the southernmost part of Japan, whereas MG005 is a winter-hardy accession from central Japan near Tokyo. 1st year is the year the seedlings were transplanted to the field site. **b** Flowering time summary for the 2014/2015 season. Gray color indicates no available data. Flowering proportion 2014 (FP 2014) was quantified as number of individuals flowering in 2014 divided by the total number of established plants. The end of flowering 2015 was quantified as the number weeks flowers were observed after senescence of the first set of pods. The first set of pods senesced in July 2015 for most accessions.

and well-established during summer and fall, but failed to recover after winter. Although it includes relatively few accessions, our reciprocal garden experiment clearly indicates that pop3 accessions are not generally more hardy in transplantation experiments and provides evidence to support that pop3 could be locally adapted to a cold climate.

We also collected phenotype data for other potentially adaptive traits. Flowering time was a likely candidate, and we

quantified flowering time at the field site in Tohoku and in a greenhouse in Denmark. *Lotus* does not require vernalization for flowering, and we did not vernalize seeds or seedlings for either experiment. In the field, we found the flowering time characteristics to depend greatly on the year and the planting date, and we analyzed data from each planting year separately. In 2014, the plants were sown on July 4th and transplanted to the field on August 4th. This late planting date caused a very strong

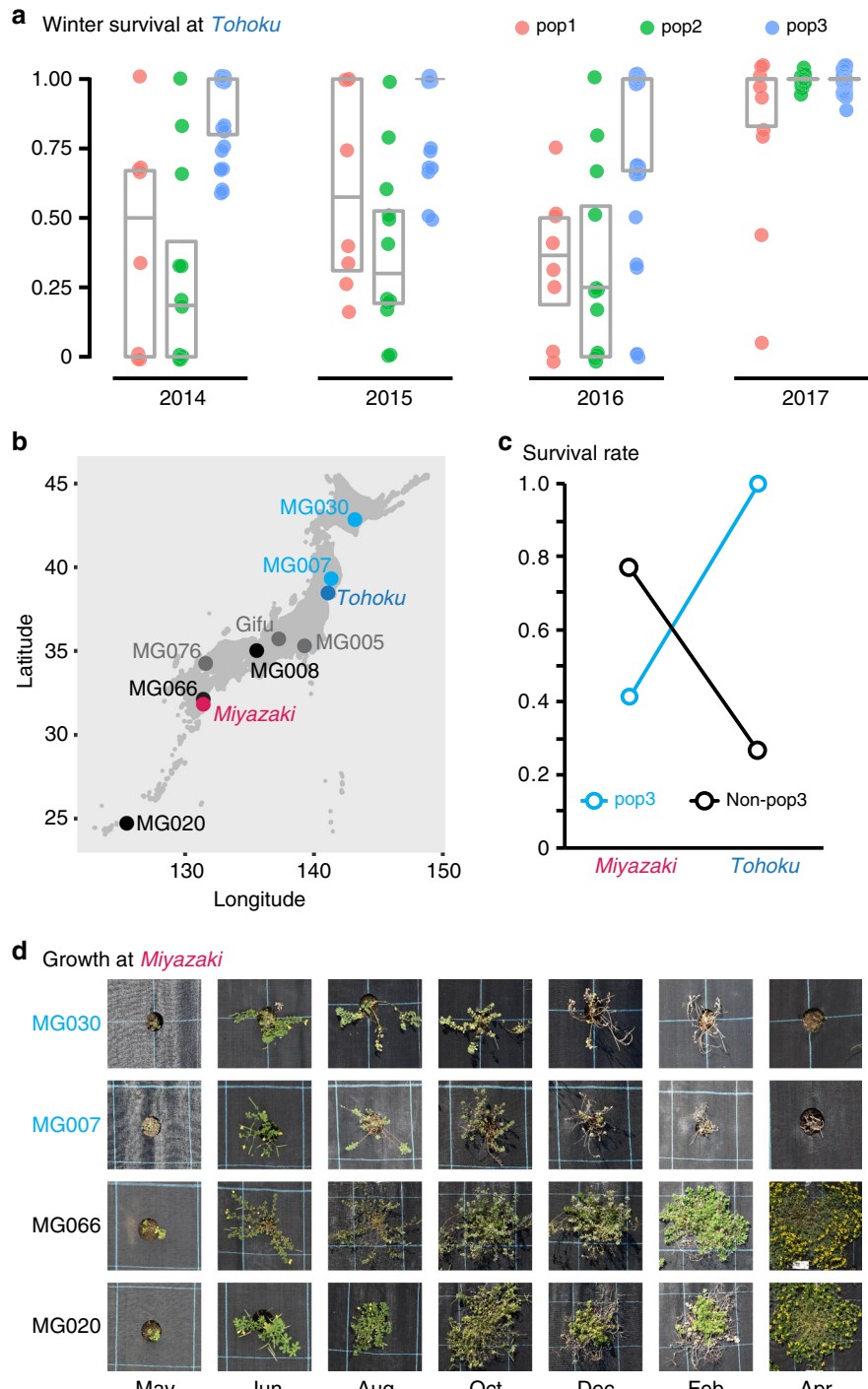

**Fig. 4 Pop3 local adaptation. a** Winter survival boxplot based on data from 610 individuals from 57 non-admixed accessions. The plots show data for the planting year indicated with each dot representing an accession mean. The horizontal gray bar indicates the median, and the lower and upper hinges correspond to the 25th and 75th percentiles. **b** The geographical origin of pop3 (blue), non-pop3 (black) and admixed (gray) accessions grown at field sites in both Miyazaki (red italics) and Tohoku (blue italics) is indicated. **c** Average survival rates of pop3 and non-pop3 accessions at the field sites in Miyazaki and Tohoku based on data from five accessions grown at both sites (MG020, MG066, MG008, MG007, and MG030). **d** Pictures from the field site at Miyazaki, 31.83°N, 131.41°E. Accessions IDs are indicated to the left and months in 2017/2018 below the pictures. Source data underlying **a**, **c** are provided as a Source Data file.

differentiation between the accessions. Many of the northern pop3 accessions failed to flower in the planting year, flowering instead the following spring (Fig. 3b). The same accessions tended also to show an early end of flowering in 2015 (Fig. 3b). Similarly to winter survival, these traits were strongly correlated with geographic origin, in contrast to the greenhouse flowering time data (Fig. 5a, Supplementary Figs. 5, 6). In the field, we also measured leaf accumulation of sodium and potassium ions, which was poorly correlated with geographic origin (Fig. 5a, Supplementary Figs. 5, 6). In addition, we quantified seed properties, which showed an intermediate correlation with geographic origin (Fig. 5a, Supplementary Figs. 5, 6).

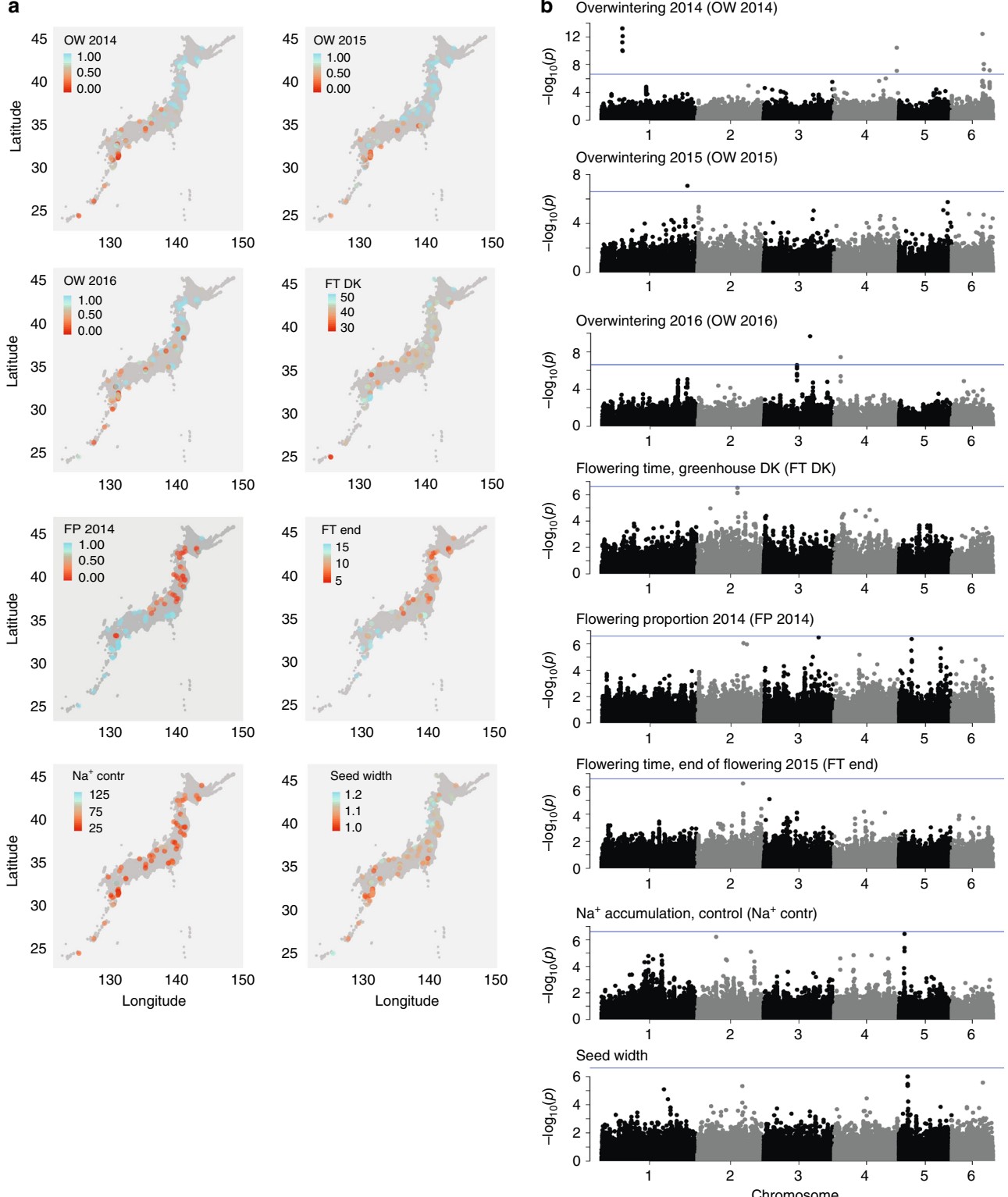

**Fig. 5 Phenotype data and GWA analysis. a** Heatmaps displaying phenotype data. OW: overwintering, the fraction of plants surviving winter. FT DK: days to flowering in a greenhouse located at 56.229406 N, 10.126424E, 8380, Trige, Denmark. FP 2014: the proportion of plants flowering at the Tohoku field site in 2014. FT end: End of flowering at the Tohoku field site in 2014/2015, quantified as the number of weeks with flowers after the first set of mature pods. Na+ contr: sodium ion accumulation in roots. Seed width: Width of seeds in mm. **b** Manhattan plots showing results of GWA scans for the phenotypes indicated. Blue line: Bonferroni threshold, $2.5 \times 10^{-7}$. Source data underlying **b** are provided as a Source Data file.

**Winter survival and pop3 differentiation**. For our population sample, linkage disequilibrium decayed to an $r^2$ value of 0.2 within ~10 kb, indicating that mapping resolution would be sufficiently high to identify a limited set of candidate genes in GWA scans (Supplementary Fig. 7). With phenotype data available for potentially adaptive traits, we proceeded with GWA analysis using a method that includes stringent correction for population structure, ensuring that significant hits represent associations that cannot be explained by the overall population structure[3,4]. We identified SNPs strongly associated with a number of the examined traits, particularly for overwintering and flowering (Fig. 5b). Overwintering 2014 displayed the most highly significant associations, which were distributed across several separate genomic loci, indicating that winter survival is a complex trait. The 57 top GWA SNPs ($p < 5 \times 10^{-5}$, GWA test) explained 48% of the phenotypic variation for overwintering 2014.

We then compared the GWA and $F_{ST}$ results to check for skews in the $F_{ST}$ value distribution for the top 200 GWA SNPs. For the traits that were strongly correlated with both geographical origin and population membership (Supplementary Fig. 5), we were concerned that population structure might cause $F_{ST}$ distribution skews unrelated to the phenotype examined, despite the correction for population structure in the GWA analysis. To investigate if this was the case, we carried out GWA analysis using permuted phenotype data for each trait. The permutation was carried out such that plant genotype and phenotype were uncoupled while ensuring that the permuted phenotypes displayed similar levels of correlation with population structure as the observed phenotypes[20]. Based on the results of 1000 permutations, we then calculated false discovery rates for the observed $F_{ST}$ skews for each trait. The SNPs associated with most flowering time traits as well as ion accumulation did not show large deviations from the $F_{ST}$ distributions derived from permuted data (Fig. 6a, b, Supplementary Fig. 8). However, SNPs associated with overwintering traits, flowering proportion in 2014, the end of flowering in 2015, and seed width showed a significant skew towards high $F_{ST}$ values (Fig. 6a, b, Supplementary Fig. 8). Moreover, it was characteristic for these traits that even among the SNPs associated with the phenotype, it was the most strongly associated SNPs that showed the highest $F_{ST}$ values, which was rarely the case for the permuted data (Fig. 6a, b, Supplementary Fig. 8).

We repeated the GWA/$F_{ST}$ overlap analysis for the pop1 vs. pop2 comparison. As expected, the SNPs associated with Tohoku overwintering traits did not show skews towards high $F_{ST}$ values for the pop1 vs. pop2 comparison (Fig. 6c, Supplementary Fig. 9). In contrast, the top SNPs associated with minimum temperature at the geographic origin of the accessions showed a very strong skew towards high $F_{ST}$ values for pop1 vs pop2 differentiation (Fig. 6c, Supplementary Fig. 9), suggesting that this, or a strongly correlated environmental variable, was likely critical for pop1/pop2 differentiation. Using the top 200 SNPs, GWA -$\log(p)$ values were anti-correlated with the $F_{ST}$ values. However, there were very few SNPs with low $F_{ST}$ values among the top 200 GWA SNPs, which meant the correlation coefficient was calculated based on a very narrow $F_{ST}$ value range. Extending the analysis to the top 400 GWA SNPs retained a low $F_{ST}$ skew false discovery rate and resulted in a positive $F_{ST}$ vs. $-\log(p)$ correlation (Fig. 6c, Supplementary Fig. 9).

As a visual illustration, we overlaid GWA and $F_{ST}$ signals for the traits and chromosomes showing the strongest GWA hits (Fig. 7, Supplementary Data 2, 3). The most striking example of overlap between GWA and $F_{ST}$ signals was found for overwintering 2014 on chromosome 6, where the top GWA hit was found in the region containing the genome-wide top $F_{ST}$ gene *Lj6g3v1790920* (Supplementary Fig. 4C), and two additional strong GWA signals overlapped with prominent $F_{ST}$ peaks (Fig. 7, **peaks 5-7**). The top 2014 overwintering GWA signal was also recovered in 2015, albeit with lower signal strength (Fig. 7, **peak 5**). The overlap between the top GWA hits for overwintering and the genome-wide top $F_{ST}$ signals indicates that winter survival has played a major role in pop3 local adaptation and points to specific candidate genes with potential adaptive roles.

Three candidate genes reside within the strongly differentiated chr6 region 20,048,000-20,068,000 comprising the top GWA overwintering signals (Fig. 7, **peak 5**). *Lj6g3v1790920* encodes a RING-type zinc finger protein, *Lj6g3v1789910* is a putative ubiquitin ligase, while *Lj6g3v1789900* is a putative 1-aminocyclopropane-1-carboxylate oxidase similar to *Arabidopsis* AT1G05010, which is involved in ethylene biosynthesis. Another prominent GWA peak on chr6 was in the region 20,961,109-20,970,404, containing a single candidate gene, *Lj6g3v1887780*, which is annotated as an HBS1-like protein containing a predicted Alpha/Beta hydrolase fold (IPR029058) (Fig. 7, **peak 6**). The protein is highly conserved across plants but has no known function. The third prominent GWA peak on chr6, 24,760,760-24,781,399 contained three candidate genes, two putative G-type lectin S-receptor-like serine/threonine-protein kinases, *Lj6g3v2130140* and *Lj6g3v2130160*, belonging to a large family of largely uncharacterized receptor-like kinases, and *Lj6g3v2130130* encoding a putative Glutamyl-tRNA(Gln) amido-transferase subunit A protein (Fig. 6, **peak 7**). On chr1, several SNPs with very strong GWA associations and high $F_{ST}$ values were dispersed across a large 240 kb region (13,458,027-13,698,337) (Fig. 7, **peak 1**). A more clearly defined peak on chr1 was located at 29,029,933-29,033,958. It contained a single candidate gene, *Lj1g3v2533770*, which encodes a protein very similar to the *Arabidopsis* FERONIA receptor-like kinase AT3G51550 (Fig. 7, **peak 2**). According to published expression data, the overwintering candidate genes were primarily expressed in roots and nodules (Supplementary Fig. 10)[21–23].

**Pop3 has diverged in flowering phenology and seed traits**. Although greenhouse flowering time produced signals in the GWA analysis (Fig. 5b), the SNPs identified did not show a pronounced skew towards high $F_{ST}$ values (Fig. 6b), and the trait was poorly correlated with geographical origin (Supplementary Figs. 5, 6). In contrast, the SNPs strongly associated with flowering proportion field data from 2014, where the accessions were planted late and showed strong differentiation, did overlap with high $F_{ST}$ scores, as did the end of flowering in 2015 (Fig. 6b). Remarkably, the top GWA hits on chromosome 2 overlapped between 2014 flowering proportion and end of flowering 2015 (Fig. 7, **peak 3**), suggesting that the same gene(s) could be involved in controlling both the onset and end of flowering. The genes, *Lj2g3v1988990* and *Lj2g3v1989150*, neighboring the top GWA hits are expressed in flowers, pods, and seeds (Supplementary Figure 10) (Høgslund et al., 2009; Mun et al., 2016; Verdier et al., 2013). *Lj2g3v1988990* is a putative mitochondrial glutamate dehydrogenase very similar to *AtGDH2* (AT5G07440), while *Lj2g3v1989150* is highly similar to *AtEMF2* (AT5G51230) and *AtVRN2* (AT4G16845), which are both flowering time regulators, with *VRN2* mediating vernalization responses in *Arabidopsis*[24,25].

Seed width, which showed less strong correlations with environmental parameters and geographical origin (Fig. 5a, Supplementary Figs. 5, 6), also showed GWA hits with a skew towards high $F_{ST}$ values, although not as extreme as for the overwintering or flowering traits (Fig. 6b). This was due to a peak on chromosome 5 (Fig. 7, **peak 4**), covering *Lj5g3v0526490*, which is a putative mitochondrial glycoprotein similar to

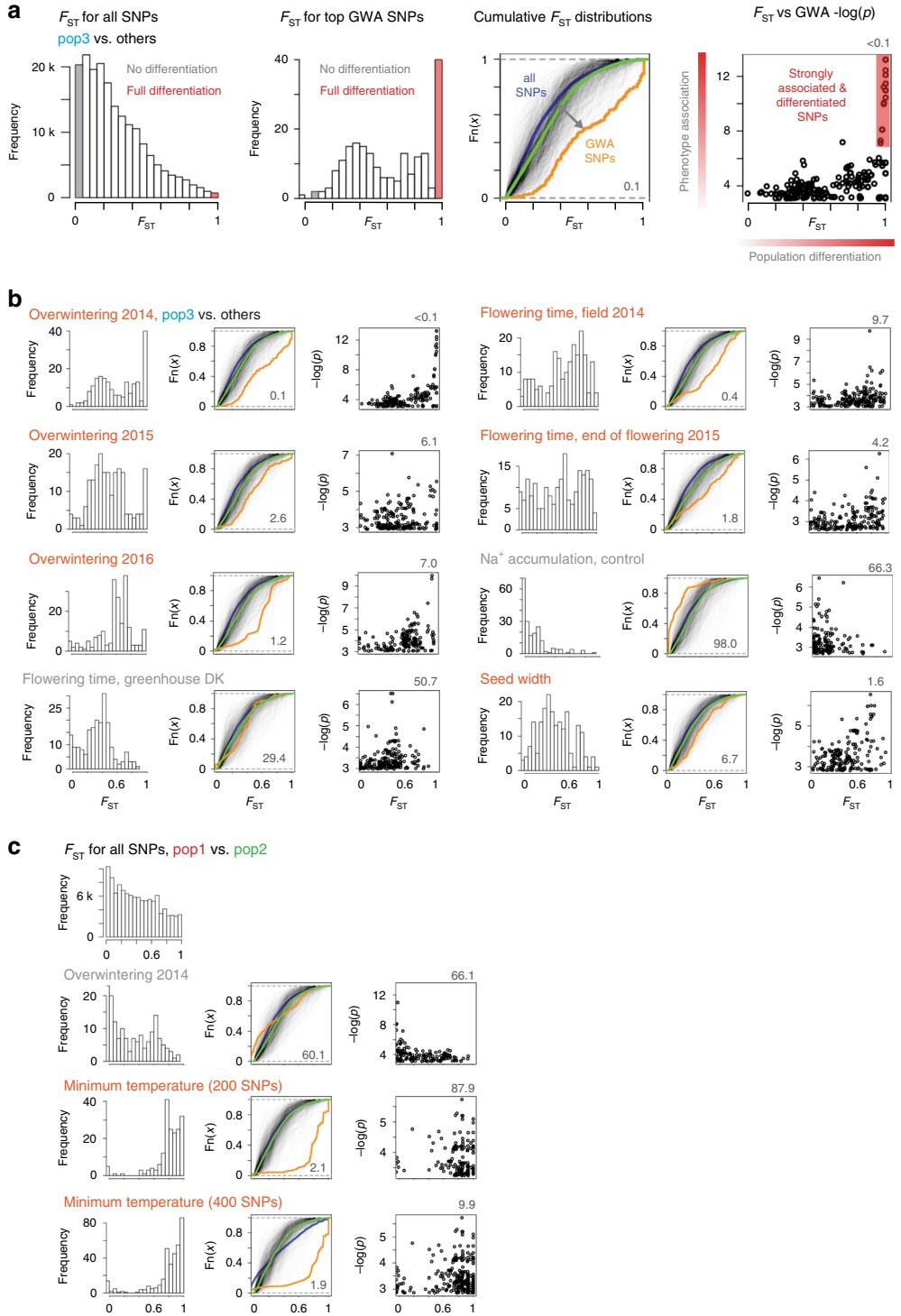

AT4G31930, and *Lj5g3v0526510* that is similar to AT5G25170. These genes with no described functions are highly expressed in seeds and flowers (Supplementary Fig. 10)[21–23].

## Discussion

Here, we have inferred the population structure and demographic history of wild *Lotus* in Japan. We conclude that *Lotus* colonized Japan from Tsushima Island after the last ice ages, reaching southern Japan before the northern part with little long-distance gene flow post colonization. It was a priori conceivable that the very different climate across Japan could be a barrier to

colonization and that selection on standing variation for phenotypic traits related to temperature and daylight would be detectable. Thus, we have exploited the *Lotus* subpopulations to study evidence of local adaptation through a combination of genotype-based $F_{ST}$ genome scans and phenotype-based GWA analyses using data from a common garden field experiment. Our study illustrates the power of this combination. Alone, $F_{ST}$ genome scans identified strong population differentiation signals, but provided no hints as to the phenotypic traits or environmental variables driving the differentiation, nor did they account for the effects of population structure. GWA analysis of phenotypic

**Fig. 6 Distribution of $F_{ST}$ values for top GWA hits. a** Reading guide for panels **b** and **c**. The leftmost panel shows the distribution of $F_{ST}$ values for all SNPs. The SNPs shaded in gray are not differentiated between the two subpopulations compared, i.e. their genotype cannot be explained by population structure. The converse is true for the SNPs shaded in red; their genotypes can be fully explained by population structure. The second panel from the left shows distributions of $F_{ST}$ values for the 200 SNPs with highest -log($p$) GWA scores for the trait indicated. The third panel from the left shows experimental cumulative distribution graphs for the $F_{ST}$ values for all SNPs (blue) and the SNPs shown in the panel to the left (orange). If the orange curve lies below the blue curve, it indicates a shift to higher $F_{ST}$ values for the SNPs with high GWA scores. Each of the semi-transparent black lines indicate $F_{ST}$ distributions for the top 200 SNPs from a GWA analysis of permuted phenotype data. The green line indicates the distribution for all 1000 sets of permuted data. The false discovery rate shown in the lower right hand corner indicates the frequency (%) of permuted distributions shifted to higher $F_{ST}$ values than the observed data. The rightmost panel shows $F_{ST}$ values plotted versus GWA -log($p$) scores. The false discovery rate shown in the upper right hand corner indicates the frequency (%) of $F_{ST}$ versus GWA -log($p$) Spearman correlation coefficients with larger values than the correlation coefficient of the observed data. **a**, **b** $F_{ST}$ values were calculated based on a pop3 vs. non-pop3 comparison. **c** $F_{ST}$ values were calculated based on a pop1 vs. pop2 comparison. Traits likely associated with local adaptation are highlighted (orange). All analyses were carried out using top 200 GWA SNPs, except for the lower panel in **c** where the top 400 SNPs were used. Source data are provided as a Source Data file.

traits, on the other hand, provided strong, population structure-corrected phenotype-genotype associations for a number of traits, but offered no way of determining which, if any, of the traits were important for local adaptation. In our study, the combination of the two approaches was powerful, because we found very pronounced skews towards high $F_{ST}$ values for certain traits, identifying these as strong candidates for driving local adaption and supplying trait labels to previously anonymous $F_{ST}$ peaks. Because of the complex effects of population structure on GWA results, permutation analysis, where the correlation between phenotype data and population structure was maintained, was important in generating trait-specific empirical data that allowed calculation of significance measures to support our conclusions.

The year-to-year differences in winter survival and their impact on the GWA results illustrates that the conditions under which the common garden experiments are carried out have a large impact on the power to detect overlapping signals from GWA and $F_{ST}$ genome scans. In our case, the top overwintering GWA signals for winter 2014 overlapped most clearly with the top $F_{ST}$ peaks. Here, the plants were transplanted late, which resulted in very strong differentiation between local and non-local accessions. Subsequent years, planting took place earlier in the year to promote plant survival and allow quantification of other phenotypic traits, resulting in less pronounced GWA signals, and when the full experiment was re-planted in 2017, the earlier planting date and milder winter resulted in survival of nearly all accessions, thwarting GWA analysis.

These observations also offer a possible explanation for why similar $F_{ST}$/GWA overlaps have not been frequently reported, since their identification might require screening several candidate phenotypic traits of putative adaptive relevance for the subpopulations under investigation. On the same note, we found it interesting that the top GWA SNPs associated with greenhouse flowering did not show a skew towards high $F_{ST}$ values, but that the SNPs associated with flowering proportion in 2014 and end of flowering of second year plants in 2015 did show pronounced skews (Fig. 5b) and pointed to a putative ortholog of an *Arabidopsis* flowering time regulator. The flowering time data again emphasized the impact of genotype by environment interactions and the need to screen multiple phenotypes in an environment relevant to the populations under study in order to identify traits and genes associated with local adaptation.

Three of the traits that showed the most pronounced $F_{ST}$ skews, Overwintering 2014, Flowering proportion 2014 and end of flowering 2015 (Fig. 6b), were all measured after transplanting the plants late in the year, forcing them to establish later than plants from natural populations in the area. These stressful conditions in 2014 caused strong phenotypic differentiation between the accessions, apparently facilitating GWA mapping of

traits associated with population differentiation. In 2017, when an earlier planting date was used that more closely approximated the phenology of natural populations, nearly all accessions survived winter. Winter survival, as quantified in 2014, might therefore not be an adaptive trait under current conditions at Tohoku, although it is possible that other factors such as pedoclimate, plant density and competition with other plant species could challenge plants in natural populations in a fashion that would reveal inter-accession differences in a similar fashion as the late planting date did in our field experiments. Alternatively, fixation of the winter hardiness alleles in pop3 accessions in northern Japan may have taken place under low temperature conditions following the last glaciation. The lower temperatures could have pushed *Lotus* germination to later in the year and made winter set in earlier during plant development. This would have reduced the time available for *Lotus* seedlings to establish before winter and imposed a stronger winter hardiness selection pressure than what is currently affecting natural populations in the area.

It may seem odd at first glance that the two flowering time traits were identified as adaptive, since many of the northern accessions did not flower in the planting year at the northern site in Tohoku. We think the lack of flowering in 2014 for northern accessions could reflect that they anticipated winter and would not flower outside of the right window of opportunity. Because of the late planting date in 2014, no accessions flowered before August, which is two to three months later than the natural populations in the region. In 2015, when the plants were well-established, they all flowered nearly synchronously in May at the same time as the natural populations. In 2015, the northern accessions showed an earlier end of flowering as compared to the southern accessions, again consistent with winter anticipation.

For a perennial plant such as *Lotus*, it makes intuitive sense that optimizing the timing of flowering and surviving harsh winters could have been critical for local adaptation to cold climates. Our work supported these hypotheses and identified the most prominent genomic regions and candidate genes associated with flowering time and overwintering. Remarkably, 48% of the phenotypic variation for winter survival 2014 could be explained by strongly differentiated and phenotype-associated SNPs, indicating that a limited number of loci with large effects have a strong impact on the trait. Such pronounced overlaps between top GWA and $F_{ST}$ signals have not, to our knowledge, been reported in the plant literature, but they are reminiscent of the characteristics displayed by SNPs associated with human skin pigmentation[7]. Our results suggest that strong selection has acted on *Lotus* alleles that are beneficial for perennial winter survival in cold climates. Based on the candidate genes we identify here, the molecular genetic mechanisms that underpin these fundamental adaptations can now be investigated.

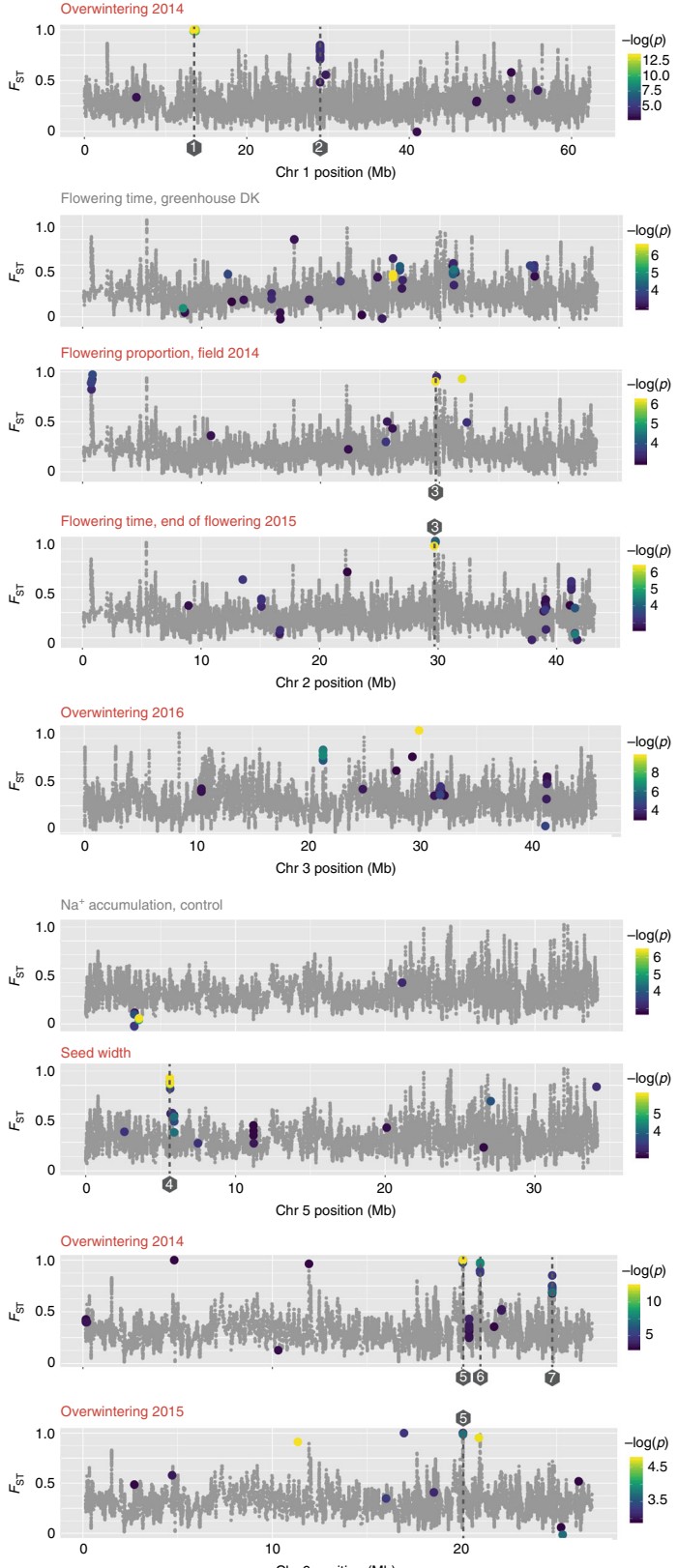

## Methods

**Read mapping and variant calling.** The wild *Lotus* accessions used in this study can be obtained from Legume Base (https://www.legumebase.brc.miyazaki-u.ac.jp). All samples were sequenced using Illumina paired-end reads (ENA accession PRJEB27969). The reads were mapped to the *L. japonicus* genome version 3.0 (http://www.kazusa.or.jp/lotus/) using Burrows-Wheeler Aligner (BWA) *mem* v0.75a with default parameters[26]. An average of 96% of the reads were mapped and

the resulting average genome coverage was 12.5 (Supplementary Data 1). Duplicates were marked in the resulting alignment (BAM) files using Picard v1.96. Then, using the Genome Analysis Tool Kit (GATK) v2.7-2 pipeline[27], reads were re-aligned in INDEL regions using the *RealignerTargetCreator* and *IndelRealigner* functions. A preliminary list of SNPs and INDELs was then generated using the re-aligned BAM files using the GATK *UnifiedGenotyper* function. Subsequently, base recalibration of the scores was done using the *BaseRecalibrator*, and the BAM files

**Fig. 7 Overlay of $F_{ST}$ values and GWA results.** $F_{ST}$ averages for 10 SNPs (gray dots) are plotted. Colored dots are shown for the 200 SNPs with highest GWA -log($p$) scores for the traits indicated. The vertical positions of the colored dots indicate their $F_{ST}$ values on the same scale as the gray dots, whereas the color indicates the GWA -log($p$) score. Numbers in gray hexagons mark the peaks discussed in the main text. 1: Chr 1, a large 240 kb region 13,458,027-13,698,337. 2: Chr 1 29,029,933-29,033,958; *Lj1g3v2533770* putative FERONIA receptor-like kinase. 3: Chr 2; *Lj2g3v1988990* putative mitochondrial glutamate dehydrogenase, *Lj2g3v1989150* similar to *AtEMF2* (*AT5G51230*) and *AtVRN2* (*AT4G16845*). 4: Chr 5; *Lj5g3v0526490* putative mitochondrial glycoprotein similar to *AT4G31930*, and *Lj5g3v0526510* that is similar to *AT5G25170*. 5: Chr 6 20,048,000-20,068,000; *Lj6g3v1790920* RING-type zinc finger protein, *Lj6g3v1789910* putative ubiquitin ligase, and *Lj6g3v1789900* putative 1-aminocyclopropane-1-carboxylate oxidase. 6: Chr 6 20,961,109-20,970,404; *Lj6g3v1887780* HBS1-like protein. 7: Chr 6 24,760,760-24,781,399; *Lj6g3v2130140* and *Lj6g3v2130160* putative G-type lectin S-receptor-like serine/threonine-protein kinases, and *Lj6g3v2130130* putative Glutamyl-tRNA(Gln) amidotransferase subunit A protein.

generated were compressed using the *ReduceReads* function followed by variant discovery, resulting in 8,716,552 putative variants that included SNPs (7,836,221) and INDELs (880,331).

The highly inbred reference accession, MG-20, was also re-sequenced. The polymorphic positions called as homozygous reference and heterozygous in MG-20 were compared to the positions with MG-20 homozygous reference calls, and the properties of these positions were analyzed. Low inbreeding coefficient and high haplotype scores characterized heterozygous positions in MG-20 (Supplementary Fig. 1). The INDEL positions had similar properties. The SNP positions were therefore required to fulfill the following criteria: Inbreeding Coefficient > 0.1, HaplotypeScore < 0.3, alternative allele quality > 30, total depth > 150 and a homozygous reference call for the MG-20 accession. Callable positions were defined as positions that fulfilled the above-mentioned criteria and had a genotype call for at least one accession in addition to MG-20.

To validate our filtering, we compared these variants to a set of independently validated SNPs genotyped using the Illumina GoldenGate system. Out of 343 validated SNPs included in the initial call set, 324 matched the genotype calls based on Gifu B-129 and MG-20 sequencing reads (Supplementary Data 4). Our stringent filtering had removed 93% of the initially called SNPs and 37% of the experimentally validated SNPs, resulting in a 9.5 fold enrichment for validated SNPs in the filtered set. Assuming that, since *Lotus* Gifu is highly inbred, all heterozygote calls in this accession would be false positives, we estimated a false positive rate of 4%. The site frequency spectrum for the high-confidence SNPs indicated that we had undercalled rare heterozygous alleles with our conservative filtering approach, most likely because these calls would only be supported by data from a small subset of the accessions (Supplementary Fig. 2).

**Calculation of genetic diversity and distance.** The --window-pi function of VCF tools version 0.1.9[28] was used to calculate genetic diversity, with the chromosome size as the window size. Pairwise genetic distances were calculated by assigning a score of +2 for homozygous differences, +1 for heterozygous differences and 0 for identical alleles at each polymorphic site and then dividing by the total number of polymorphic sites. Geographical distances were calculated using the distVincentyEllipsoid function from the R package geosphere (see https://github.com/ShahNiraj/JapanHistory).

**Principal component and linkage disequilibrium analysis.** PCA analysis was performed with EIGENSOFT v6.0beta[29,30] using the *smartpca* command with default parameters. PCA plots were generated using R version 3.4.3. Linkage disequilibrium (LD) was calculated using VCFtools v0.1.9[28] for all the SNP-pairs with a maximum physical separation of 50,000 base pairs. Chromosome 0 was ignored in the analysis. A randomly selected set of 200,000 SNP-pairs was plotted using R version 3.4.3.

**Population structure analysis.** To infer population structure, a Bayesian model based clustering analysis was conducted with fastSTRUCTURE version 1.0[19] for 201,694 non-repetitive SNP markers with minor allele count > 5 in 136 accessions. The analysis was run with a number of clusters ($K$) ranging from 1 to 8 with default parameter settings.

**Calculation of selfing rate.** In order to estimate the average selfing rate of the *Lotus* accessions investigated in this study, the average expected and observed heterozygosity over all loci were calculated for each subpopulation separately using the R-package 'adegenet'[31]. These values were used to calculate Wright's inbreeding coefficient ($F_{IS}$) for each population given that $F_{IS} = 1 − $ (observed heterozygosity/expected heterozygosity). The average selfing rate (s) was then calculated as Eq. (1)

$$s = 2 * F_{IS}/(1 + F_{IS}) \qquad (1)$$

where $F_{IS}$ is the average inbreeding coefficient of all 3 populations.

**PSMC analysisn.** For estimation of population size history, the Pairwise Sequentially Markovian Coalescent (PSMC) model was used[17]. Accessions with more than ×10 coverage were subsampled to ×10 coverage. Diploid consensus sequences were made with SAMtools v1.3 and BCFtools v1.3 and SNPs which had

mapping quality > 50 and read depth between 5 and 100 were included in the analysis. PSMC analysis was run with each pseudodiploid consensus sequence using the recommended settings[17]. The mutation rate per nucleotide in *A. thaliana*, $6.5 \times 10^{−9}$ year/site[18] was used in the visualization step.

**Phenotyping.** To investigate the plant phenotypes under field conditions, we grew the wild *Lotus* accessions in a field at Tohoku, Graduate School of Life Sciences, Tohoku University (38.46°N, 141.09°E) located in Miyagi Prefecture, Japan from 2014. Partly scrubbed seeds were sown on cell trays containing a 1:1 mixture of soil and vermiculite, and grown in a greenhouse for around one month before transplanting to the field. Sowing and transplanting dates were 3rd of July and 4th of August in 2014, 30th of May and 30th of June in 2015, 3rd of June and 13th of July in 2016, and 30th of March and 28th of April in 2017, respectively. From 2014 to 2017, a total of 106 wild accessions were grown in the same field, including overwintered individuals in 2015 to 2017 seasons. The accessions were subdivided into three groups, each covering a wide range of geographic origins. Within each of these groups, the accessions were planted ordered by their accession ID in the column direction in the field. Three individuals from the same accession were planted spaced least 30 cm apart directly in field soil. Subsequent sets of three individuals per accession were planted at 60 cm intervals. Each accession was planted in two sets of three, a total of six plants per experiment, in control and salt stressed fields. Salt stress treatments were conducted by irrigation with underground water containing salt around 1/4 concentration of seawater and normal water (Control). Leaf samples from three plants of each line were taken after four weeks of treatment for measurement of ion contents (Na$^+$ and K$^+$) in 2014. In the 2017 season, a new field was used for growing 136 wild accessions in control field only. To carry out comparative phenotype analysis in multiple field, we grew eight *Lotus* accessions in a field at University of Miyazaki (31.83°N, 131.41°E) in Miyazaki Prefecture, Japan in the 2017–18 season. Sowing and transplanting dates were 26th of April and 24th of May in 2017, respectively. The field design was same as that of Tohoku University field.

Greenhouse flowering time was quantified as days to the first open flower in a greenhouse located at 56.229406 N, 10.126424E, 8380, Trige, Denmark. Seeds were germinated on wet filter paper and seedlings were transplanted to the greenhouse on May 10th, 2014, where they were grown in peat (Pindstrup, mix 2) mixed with 20% perlite at 18–23 °C (day) 15 °C (night) using a 16/8 hour day/night cycle and 70% relative humidity fertilized with Pioner NPK Makro 14-3-23 + Mg blå.

For seed phenotyping, seeds were collected from the plants grown in a greenhouse (Trige Aarhus, Denmark) in 2014. Seed images were obtained using Epson Perfection 600 Photo scanners. The image files were analyzed using SmartGrain[32] to count the total number of seeds and to measure the area, size, length, width, length-width ratio, and circularity of each seed.

**Statistical analysis of winter survival.** Differences in winter survival as a function of location and population membership were evaluated using linear mixed model analysis with the R package lme4[33]. The planting scheme, where each accession was grown in multiple blocks of three individuals, resulted in a nested structure of the experimental data with Block nested within Accession nested within Population. Taking the nested structure into account, we evaluated the relative effects of the field setup and the population membership using the following model: 'lmer(survival ~ (1| Population) + (1|Field.row) + (1|Field.Column) + (1|Population:Accession) + (1| Population:Accession:Block)' and included the nested Accession and Block terms in the analysis of population 'lmer(Survival ~ Population + (1|Population:Accession) + (1|Population:Accession:Block))' and location 'lmer(Survival ~ Location + (1|Population:Accession) + (1|Population:Accession:Block))' effects. Statistical significance was then estimated based on the resulting models using the lmerTest and multcomp packages 'glht(model, linfct = mcp(Population = "Tukey"))'[34]. Population by Location interaction effect was evaluated using the model 'lmer(survival ~ Population + Location + Location:Population + (1|Population:Accession) + (1|Population:Accession:Block))' followed by ANOVA analysis 'anova(model)'.

**$F_{ST}$ and GWA analysis.** The --weir-fst-pop function from VCF tools version 0.1.9[28] was used to calculate fixation indices ($F_{ST}$). Prior to GWA analysis, missing genotype data points were imputed using Beagle version 5.0[35], and the VCF file was converted to a simple genotype-only format. GWA scans were carried out using a

python implementation of EMMA (https://app.assembla.com/spaces/atgwas/git/source)[3,4] with a minor allele count cutoff of 8. The EMMAX algorithm was run for all SNPs followed by p-value adjustment using EMMA for the top 200 SNPs. The software GCTA-GREML was used to estimate the proportion of phenotypic variance explained by significant SNPs. For selected traits, the method was applied to capture variance explained by a subset of associated SNPs GWA p-values ≤ 5e^−5. SNPs with a minor allele count below 10 were not included.

**Permutation analysis and false discovery rate calculations**. One thousand permutations of the trait values with respect to their accession IDs were carried out for each trait and used for GWA analysis (https://github.com/bvilhjal/mixmogam/blob/lotus/examples.py). The simulation strategy accounted for population structure by sampling phenotypes with a given covariance, where the covariance matrix was set to the estimated covariance matrix ($\sigma_g^2 K + \sigma_e^2 I$) obtained from the linear mixed model fit on the observed trait. This sampling strategy ensures that the sampled phenotypes display similar levels of correlations with population structure as the observed phenotype, thus reducing potential of bias due to population structure. False discovery rates for $F_{ST}$ skews were calculated by running Kolmogorov–Smirnov tests comparing the $F_{ST}$ distributions of each set of top 200 SNPs from the permuted GWA runs to the top 200 SNPs from the observed data. We then counted how many times the Kolmogorov–Smirnov tests estimated at least 1000-fold difference between the probability that the observed $F_{ST}$ distribution was above or below the permuted $F_{ST}$ distribution and noted the direction of the significant skews. For example, for Overwintering 2014 we found that 992 times the observed $F_{ST}$ distribution was skewed towards higher $F_{ST}$ values compared to the permuted sets, 1 time the observed $F_{ST}$ distribution was skewed towards lower $F_{ST}$ values, and seven times there were no significant skews, yielding an fdr of 0.001 [1/(1 + 992)]. To calculate fdr values for $F_{ST}$ vs. −log(p) correlations, we calculated the Spearman correlation for each permuted set and counted how many values were above and below, respectively the observed value. For example, for overwintering 2014, we found 1000 Spearman correlation values below the observed value and 0 above, yielding an fdr of less than 1/1000.

**Reporting summary**. Further information on research design is available in the Nature Research Reporting Summary linked to this article.

## Data availability
Data supporting the findings of this work are available within the paper and its Supplementary Information files. A reporting summary for this Article is available as a Supplementary Information file. The datasets generated and analyzed during the current study are available from the corresponding author upon request. Accessions read data have been deposited in the European Nucleotide Archive (ENA) database with accession number PRJEB27969 [https://www.ebi.ac.uk/ena/data/view/PRJEB27969]. Genotype data is available for online GWA analysis through a website (https://lotus.au.dk/gwas/) based on the GWAPP platform[36], where it can be selected during the Genotype step when creating a new GWA analysis. The phenotype data and GWA analyses are also available from the site (https://lotus.au.dk/gwas/#/study/68/overview). The source data underlying Figs. 1d, e, 4a, c, 5b, 6, as well as Supplementary Figs. 1, 5, 6, 8, 9 are provided as a Source Data file.

## Code availability
Custom scripts and workflows are freely available on GitHub in the repositories https://github.com/ShahNiraj/JapanHistory, https://github.com/cks2903/Lotus_data_2019, and https://github.com/bvilhjal/mixmogam/blob/lotus/examples.py.

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

## Acknowledgements
The work was supported by the Danish National Research Foundation grant DNRF79 (JS), the Genome Information Upgrading Program of the National BioResource Project in 2014 (SS), a JST CREST grant (number JPMJCR16O1) (SS), grant no 6108-00385 A from The Danish Council for Independent Research | Natural Sciences (MHS) and grant

no. 10-081677 from The Danish Council for Independent Research | Technology and Production Sciences (SUA). Wild accessions of *L. japonicus* were provided by the National BioResource Project 'Lotus/Glycine'.

## Author contribution

Conceptualization, S.U.A., S.S. and M.H.S.; Methodology, S.U.A., M.H.S. and S.S.; Software, Ü.S.; Validation, N.Sh., S.U.A., M.H.S. and S.S.; Formal Analysis, N.Sh., T.W., C.K.S., S.U.A., V.G., B.V., E.F., F.A. and K.S.; Investigation, Y.Ka., M.-Z.W., M.S., Y.I., H.J., T.M., N.Sa., S.K., Y.Ki., S.N., T.K., M.S., Y.M., M.H., H.T. and A.H.; Resources, J.S., S.S., M.H.S., S.U.A., K.L.N. and R.A.; Data Curation, N.Sh.; Writing – Original Draft, S.U.A. and N.Sh.; Writing – Review & Editing, S.U.A. and M.H.S.; Visualization, N.Sh., T.W., S.U.A. and C.K.S.; Supervision, S.U.A., S.S. and M.H.S. Project Administration, J.S., S.S. and S.U.A.; Funding Acquisition, J.S., S.S., S.U.A., M.H.S. and R.A.

## Competing interests

The authors declare no competing interests.
