## [Peer Review File · Nature Communications]

Reviewers' comments:

Reviewer #1 (Remarks to the Author):

In this study, the authors seek to reconstruct the colonization history of the perennial herb *Lotus japonicus* in Japan and to identify genomic regions and candidate genes responsible for local adaptation. The study is based on sequencing data for a large collection of accessions from across Japan, and phenotyping of plants in a common-garden established in the northern part of the range, and in a greenhouse experiment. Central is a comparison of results from a GWA analysis corrected for population structure and *Fst*-outlier detection. The results are interesting as they illustrate an approach to connect *Fst*-outliers to trait differentiation and putative adaptive differentiation among populations.

My main comments regard the design and interpretation of the field experiment. First, the authors found that the northern subpopulation had higher survival than had the two other subpopulations when planted at a site in northern Japan, and suggest that this provides evidence of local adaptation. It is true that this result is consistent with local adaptation, but a true test of local adaptation would require that similar experiments were conducted in the regions of the other two subpopulations and that the local population consistently outperformed the non-local populations (rather than pop3 having the highest fitness in all regions). I therefore suggest that statements that "pop3 is locally adapted" should be moderated (see, for example, p. 12, third paragraph, final sentence). Without additional data, it cannot be excluded that pop3 would outperform the other two subpopulations in common-garden experiments also in other regions of Japan. Second, field plantings were repeated several years but with very different planting dates (planting date varying between 28 April and 4 August). With which planting date were transplanted individuals most similar in development and phenology to those of plants in natural populations of this species in the area? For fitness variation observed in the common-garden to be relevant for the interpretation of fitness differences among natural populations, conditions in the garden and plant phenotypes should mimic those of the natural populations as closely as possible.

It is interesting that the strength of selection apparently varies among years (cf. p. 15, second paragraph). To illustrate this, I suggest the authors include survival data also from the 2017 experiment in Fig. 3B.

I suggest the authors reorganize the Introduction such that the study system is introduced after the conceptual framework has been presented.

Some of the wording in the Abstract is vague and should be clarified: (a) "recent colonization". Please, indicate approximately when colonization occurred. (b) "We carefully track the colonization process" – More correct to say that the colonization process was inferred from patterns of genomic variation? (c) "Contrasting these analyses"- What does this mean?

p. 4, line 2. Is the study species diploid?

p. 4, line 5, "is being characterized". Change to "has been examined"?

p. 4, second paragraph, first sentence. I would argue that the key comparison in reciprocal transplants is that between local and non-local populations at two or more sites. Moreover, common-garden experiments are not necessarily conducted at the sites of natural populations, which means that there is not always the possibility to compare "locals and immigrants" in such experiments.

p. 4, second paragraph, third sentence, "Such experiments...". Please, provide references to this and the following two sentences. In addition, what is meant by "molecular events"?

p. 4, second paragraph, fourth sentence. Why "subpopulation" rather than "population"?

p. 6, first paragraph, line 4. Not clear why human dispersal should prevent the identification of traits and alleles of adaptive significance.

p. 6, second paragraph, final sentence. I suggest the authors moderate this statement since no direct test of local adaptation in reciprocal transplant experiments is available.

p. 10, second paragraph, final sentence. Without any data from common-garden experiments in the regions occupied by the other populations, this does not constitute any clear evidence of local adaptation. For this to constitute evidence of local adaptation would require that fitness differences were reversed in common-gardens in the other regions.

p. 6, second paragraph, final sentence. Change "by" to "during"?

p. 7, second paragraph, third sentence, "self-compatible". And highly selfing?

p. 9, second paragraph, final sentence, "migrated north from southern Japan". But the initial spread was towards the south if the original point of colonization was Tsushima Island as proposed on the previous page?

p. 10, first line. Does region refer to the extent of the gene, and "alternative allele" to alleles of the 20 SNP mentioned? Please, clarify.

p. 10, third paragraph. What is the flowering time of natural populations in the region of the experiment and how does that compare to the flowering times recorded in the field experiment? The description of the flowering time results needs to be clarified. From the text one can get the impression that the analysis of flowering time in the field confounded variation in flowering time (within a given flowering season) and year of first flowering, whereas correlations presented in the supplement suggest that flowering times recorded in different years (i.e., of plants of different age(?)) were analysed separately, as would be appropriate. Please, clarify! Moreover, it is not clear what "this trait" on the penultimate line of p. 10 refers to.

p. 10, third paragraph. "Many of the northern accessions failed to flower in the planting year". Does this imply that flowering the first year was maladaptive? This also suggest that the GWA of flowering time in the first year was based on data for fewer accessions compared to GWA of flowering time in subsequent years? May this have affected the results? Please, clarify.

p. 19. The authors should provide more details regarding experimental design and analysis: Seeds were planted in the field in four consecutive years. Was the analysis of flowering time in different years conducted separately for plants of different age? From the description it seems that, in at least some years, plants of different age were grown together. What were sample sizes per accession in different years? How was flowering duration defined?

Graphs such as Fig. 3C are nice because they give a broad overview of geographical variation, but they should be complemented with tables giving means, SD, and sample sizes for individual accessions as supplemental material.

p. 11. What was flowering time in the field experiment in 2014 compared to that of natural populations in the area? The authors suggest that the flowering time in the field experiment that year was mainly a consequence of the late planting. This suggests that this trait may represent ability to rapidly reach reproductive maturity rather than flowering time of reproductive individuals. This should be considered when interpreting the results.

p. 11, final paragraph. Aren't high F_{st} expected for top SNP associated with traits that differ between "subpopulations"? Can this be considered an "independent test of local adaptation" when GWA and F_{st} analyses are based on the same set of accessions (see also Discussion first paragraph, "the power of this combination")?

p. 12, second paragraph. For the same reason, because pop1 and pop2 did not differ markedly in overwintering survival, it shouldn't be surprising that F_{st} values for overwintering SNP in the comparison between pop1 and pop2 were not high. Or am I missing something here?

p. 12, second paragraph, final sentence, "this environmental variable was likely critical...". Or some other environmental factor correlated with minimum temperature!

p. 15, second paragraph, second sentence. Move "for winter 2014" to after "GWA signals"?

p.19, first paragraph. How are the 136 accessions distributed across sampling sites ("local populations")? Were all from distinct sites or did some local populations contribute more than one accession?

p. 20, first paragraph. The authors state that 136 accessions were planted in 2017. How many accessions were included in the field experiments established in each of the preceding years?

p. 20, second paragraph. Does this species need vernalisation for flowering? Does vernalisation requirement vary among accessions as is the case in other species? Did the greenhouse plants receive any stratification/vernalisation treatment as would most likely plants in natural populations?

Reviewer #2 (Remarks to the Author):

This manuscript studied the local adaptation of *Lotus japonicas* in Japan. Firstly, authors resequenced 136 accessions, and performed detailed analyses of population genetics. Secondly, authors phenotyped these accessions in field and also in the growth house. Lastly, they made GWAS analysis based on different traits that could be correlated with adaptation, and looked at the overlap between highly differentiated regions (F_{st}) and GWAS signals. In general, this is a very good study about local adaptation. However, I have some concerns and comments about the manuscript below.

Major:

1) In Figure 1 and 2, it would be better to reorganize the two figures, since both shows the results of population structure and differentiation, but not well-organized. We studied the population structure of the populations firstly (mostly Figure 2C), then based on results we could divide samples into different populations.

2) PSMC analysis will be robust reveal more early evolution history than MSMC, it is better to add the MSMC analysis as comparison to reveal the demographical history of *Lotus japonicas*. This is highly rewarding given *Lotus japonicas* distributed to Japan very recently.

3) For the analyses of F_{st} and GWAS, the cut-offs are not clear. What is the rationale to use a cut-off of p-value $\leq 5e-5$ for GWAS? Usually people calculate the cut-off as $0.01/\text{total SNPs}$. For F_{st} analysis, authors mentioned $F_{st} > 0.65$, but what is the rationale behind this value as a cut-off. Given both of these analyses are fundamental results for the main conclusion of this study, it is important to carefully describe these methods.

4) Although this study did not perform the functional validation like forward genetics, it is helpful to discuss the candidate genes in more detail, based on the function of orthologous gene in *Arabidopsis* or other well-studied plants.

5) In the results section the third paragraph, "tight clustering ...", it would be better to make the claim based on π value at the genome level, actually authors also mentioned π value in Figure 2E.

Minor:

- 1) In Figure 1, it is better to explain what are the meaning of the different colors.
- 2) In Figure 1D-E, samples MG110 and MG111 were not marked on the figure, it is hard to recognize and follow here.
- 3) The order of the citation of Figures need to be updated, Figure 4 shows up in the main text early than Figure 3, authors need proofread and correct these kinds of problems.
- 4) It is better to mark all the genes mentioned in the main text that are overlapped between Fst and GWAS on the figure 5.
- 5) It takes much more effort to understand the figure 4, it is better to explain the figure in the figure legends in detail. Also in Figure 5, the meaning of dots in different colors should be explained clearly.

Reviewer #3 (Remarks to the Author):

Shah et al. study the process of the historical colonization and range expansion in *Lotus japonicas* in Japan using a combination of population genomics and common garden studies. The project is motivated by an interest in the genes impacting local adaptation. Can they be discovered and their evolutionary history explored? The project relies on the collection of a representative accession panel and shallow resequencing of this material, along with studies of population structure and trait associations. Together, these data build a picture of the timing and order of range expansion and suggest a number of potential environmental factors and plant traits impacting adaptation. Overall, this is a nice contribution within the context of the study system.

The project centers on using Fst measures of differentiation and coalescent analyses to understand population structure and genetic ancestry of population. Overall, the analyses identify several strongly differentiated groups (median Fst \sim 0.25) and infer that Kyushu Island is the center of diversity and origin of colonization of *L. japonicas* in Japan. The manuscript discusses some divergence times and builds a plausible demographic history of these populations. The manuscript then focuses on discussing genes with extreme Fst statistics as likely candidates underlying divergent selection and leading to adaptation between subdivided populations. This is a relatively common approach that can be informative for generating hypothesis. However, it should be acknowledged that there will always be outliers and the tail of the neutral Fst distribution can be remarkably long (high Fst by chance alone; see various papers by Mike Whitlock for detailed discussion). The manuscript provides no real guidance as to what Fst values might be expected under purely neutral processes.

The novel aspect of this manuscript is that the accession panel was planted in common garden in the field and that a series of phenotypes thought to be important in adaptation were screened. As such, the group can ask whether loci with effects on candidate traits are also highly differentiated. These data are nicely presented in Figure 4. The inference is that the joining of these data lead greater credence in identifying candidate genes and their role in local adaptation and subsequent differentiation. I think this is so, but only up to a point.

One major limitation in this endeavor is the potential confounding effects of population structure. GWAS is premised on the assumption that associations between marker and traits are caused by physical linkage generating linkage disequilibrium (LD). Unfortunately, many other evolutionary processes can also impact allele frequencies and LD causing spurious associations between markers and traits. The best scenario for GWAS is therefore in a randomly mating/panmictic population as here the major driver of LD is physical distance. In this case, observed correlations between markers/traits are likely to lead you to causal genes. However, most systems have some degree of population structure. Depending on the severity of population structure, some false positive associations are likely to occur. The problem is sorting out which hits are false positives versus real causal genetic signal. The problem can be really severe if the traits of interest are differentiated in the same direction as global population structure (as is the case here). In this case, any polymorphism that shows allele

frequency differences by population structure will show an association with the similarly differentiated trait.

GWAS analyses often include population structure as a covariate in the model. However, we are never able to fully “control” for population structure, and in some cases including it in the analyses can actually reduce the ability to see real signal. The authors I’m sure are well aware of these limitations. However, I felt like much of the text is written as if these are minor problems that are adequately addressed by including kinship in the analyses. The same problem holds for looking at the joint pattern of Fst outlier and GWAS signal. I think the manuscript needs to carefully address this limitation.

I have a number of other specific comments:

The manuscript mentions a number of divergence/cross coalescence dates but never gives a sense for the uncertainty in these estimates. Is it possible to put confidence intervals or bounds to these estimates?

The manuscript assumes the reader knows a bit about Lotus natural history. It might be good to include a paragraph (at least in a supplement) about the mating system, life history, and basic ecology of Lotus.

The Emma model used for association mapping is pretty conventional – however, there are other approaches that could be considered (Q + K, PCAs). Given the strong population structure in the system it would be good to justify the basic Emma kinship model is best. Perhaps some comparisons could be included in a supplement.

As far as I can tell, none of the statistical tests are controlled for multiple testing. Only nominal p-values are provided. The authors should at least address this concern and provide their philosophy for interpreting genomewide p-values.

One way to get a feel for the degree of population confounding is simply to ask how well population structure alone can predict phenotypes. This can be accomplished by predicting phenotype from population membership coefficients, for example. The best case scenario would be no relationship...

The manuscript mentions “enrichments” in several contexts – however, p-values are never presented in the text for those enrichments. Can randomization tests be used to explore the overlap of GWAS LOD and Fst outlier statistics? It might be nice to see how often these associations can arise by chance.

Phenotypes were collected from common gardens over multiple years. However, it seems each year is analyzed separately. It may be that power could be increased if these data are joined in a mixed model. Mortality from the first year is interpreted as freezing tolerance and overwintering survivorship. However, it could also be related to transplant shock and general vigor. Is there any way to disentangle these alternatives?

My understanding is some of the plant material died across successive common garden years. One of the ideas is that this loss represents selection from freezing susceptibility (it’s not random). As such, doesn’t this selection potentially cause problems in the GWAS analysis as it may be generating strong LD? The manuscript mentions some replanting.....but the details weren’t clear to me.

The discussion section mentions evidence for “tradeoffs” I don’t think the manuscript currently provide any insight or evidence for tradeoffs and how they’d maintain variation.

The details of the experimental design of the common gardens were not clear. The methods section mentions planting 3 seedlings per pot and then each line being represented by two “spots”. So, is the

pot the unit of replication?

The salt stress experiment is not really mentioned or explored in the text – this seemed a bit odd.

Point-by-point response to reviewer comments

We wish to thank the reviewers for their very careful evaluation of our work and for the detailed and constructive comments that helped us to produce a substantially improved revised manuscript.

Reviewer #1 (Remarks to the Author):

*In this study, the authors seek to reconstruct the colonization history of the perennial herb *Lotus japonicus* in Japan and to identify genomic regions and candidate genes responsible for local adaptation. The study is based on sequencing data for a large collection of accessions from across Japan, and phenotyping of plants in a common-garden established in the northern part of the range, and in a greenhouse experiment. Central is a comparison of results from a GWA analysis corrected for population structure and *Fst*-outlier detection. The results are interesting as they illustrate an approach to connect *Fst*-outliers to trait differentiation and putative adaptive differentiation among populations.*

My main comments regard the design and interpretation of the field experiment.

1.1) First, the authors found that the northern subpopulation had higher survival than had the two other subpopulations when planted at a site in northern Japan, and suggest that this provides evidence of local adaptation. It is true that this result is consistent with local adaptation, but a true test of local adaptation would require that similar experiments were conducted in the regions of the other two subpopulations and that the local population consistently outperformed the non-local populations (rather than pop3 having the highest fitness in all regions). I therefore suggest that statements that “pop3 is locally adapted” should be moderated (see, for example, p. 12, third paragraph, final sentence). Without additional data, it cannot be excluded that pop3 would outperform the other two subpopulations in common-garden experiments also in other regions of Japan.

Response to 1.1)

We have modified the statements as recommended, i.e. “Whereas pop3 showed higher winter survival rates than pop1 and pop2 at the field site near Tohoku, ...” and “This is consistent with pop3 local adaptation, although we cannot rule out that pop3 would outperform pop1 and pop2 in other environments as well.”

1.2) Second, field plantings were repeated several years but with very different planting dates (planting date varying between 28 April and 4 August). With which planting date were transplanted individuals most similar in development and phenology to those of plants in natural populations of this species in the area? For

fitness variation observed in the common-garden to be relevant for the interpretation of fitness differences among natural populations, conditions in the garden and plant phenotypes should mimic those of the natural populations as closely as possible.

Response to 1.2)

This is an interesting point, and we have now included additional figures and information to illustrate the seasonal variation in Lotus growth. Generally, the earlier planting dates are more similar to the natural populations in the area, but these induce little differentiation between the accessions, suggesting that exposing the plants to stressful conditions is required to emphasize the adaptations and genetic differences that facilitated colonisation of northern Japan. We have discussed this and added a new Figure 3 containing pictures of field-grown lotus plants and an illustration seasonal shifts between vegetative and reproductive growth.

1.3) It is interesting that the strength of selection apparently varies among years (cf. p. 15, second paragraph). To illustrate this, I suggest the authors include survival data also from the 2017 experiment in Fig. 3B.

Response to 1.3)

We have added survival data from 2017 as requested.

1.4) I suggest the authors reorganize the Introduction such that the study system is introduced after the conceptual framework has been presented.

Response to 1.4)

We have re-organized the introduction as suggested.

1.5) Some of the wording in the Abstract is vague and should be clarified: (a) "recent colonization". Please, indicate approximately when colonization occurred. (b) "We carefully track the colonization process" – More correct to say that the colonization process was inferred from patterns of genomic variation? (c) "Contrasting these analyses"- What does this mean?

Response to 1.5)

The abstract has been re-written to clarify the points mentioned.

1.6) p. 4, line 2. *Is the study species diploid?*

Response to 1.6)

Yes, Lotus is diploid, and we have now mentioned this in the introduction.

1.7) p. 4, line 5, *“is being characterized”*. Change to *“has been examined”*?

Response to 1.7)

We made the suggested change.

1.8) p. 4, second paragraph, first sentence. *I would argue that the key comparison in reciprocal transplants is that between local and non-local populations at two or more sites. Moreover, common-garden experiments are not necessarily conducted at the sites of natural populations, which means that there is not always the possibility to compare “locals and immigrants” in such experiments.*

Response to 1.8)

That is a good point. We have removed the statement “, with the local vs. immigrant comparison arguably offering stronger evidence”.

1.9) p. 4, second paragraph, third sentence, *“Such experiments...”*. Please, provide references to this and the following two sentences. In addition, what is meant by *“molecular events”*?

Response to 1.9)

We have added the reference Fournier-Level et al., 2011 as an example, and more examples are described later on in the introduction. We have changed the sentence including “molecular events” to “... to begin understanding the genetics of local adaptation”.

1.10) p. 4, second paragraph, fourth sentence. Why *“subpopulation”* rather than *“population”*?

Response to 1.10)

Subpopulation was changed to population.

1.11) p. 6, first paragraph, line 4. Not clear why human dispersal should prevent the identification of traits and alleles of adaptive significance.

Response to 1.11)

This is because human dispersal uncouples the plant genotype from its true geographical origin, which complicates interpretation of analyses of local adaptation.

1.12) p. 6, second paragraph, final sentence. I suggest the authors moderate this statement since no direct test of local adaptation in reciprocal transplant experiments is available.

Response to 1.12)

We have modified the statement so it now reads "... and show evidence for adaptation to a cold climate in a common garden experiment".

1.13) p. 10, second paragraph, final sentence. Without any data from common-garden experiments in the regions occupied by the other populations, this does not constitute any clear evidence of local adaptation. For this to constitute evidence of local adaptation would require that fitness differences were reversed in common-gardens in the other regions.

Response to 1.13)

We have modified the statement so that it now reads: "This is consistent with pop3 local adaptation, although we cannot rule out that pop3 would outperform pop1 and pop2 in other environments as well."

1.14) p. 6, second paragraph, final sentence. Change "by" to "during"?

Response to 1.14)

We have modified the statement to "... , allowing us to identify traits and associated genomic loci that were direct targets of selection during local adaptation."

1.15) p. 7, second paragraph, third sentence, “self-compatible”. And highly selfing?

Response to 1.15)

We have estimated the selfing rate to 94% based on the heterozygosity rates $s=2F_{IS}/(1+F_{IS})$, where $F_{IS}=1-\text{obs}(\text{heterozygosity})/\text{exp}(\text{heterozygosity})$, where the expected heterozygosity is calculated from the allele frequencies in the local population. The selfing rate is now stated in the manuscript.

1.16) p. 9, second paragraph, final sentence, “migrated north from southern Japan”. But the initial spread was towards the south if the original point of colonization was Tsushima Island as proposed on the previous page?

Response to 1.16)

Yes, we have modified the statement so that it now reads: “...consistent with the hypothesis that Lotus first migrated south from a starting point near Tsushima Island and then later colonized the north of Japan, losing genetic diversity along the way.”

1.17) p. 10, first line. Does region refer to the extent of the gene, and “alternative allele” to alleles of the 20 SNP mentioned? Please, clarify.

Response to 1.17)

To clarify, we have modified the sentence so it now reads “The top gene Lj6g3v1790920 spanned 1.2 kb and had a mean F_{ST} value of 0.99 across 20 SNPs. It was located within a ~20 kb genomic region showing very strong fixation of the alternative allele in pop3 relative to accessions without pop3 membership (Supplemental figure 4C).”

1.18) p. 10, third paragraph. What is the flowering time of natural populations in the region of the experiment and how does that compare to the flowering times recorded in the field experiment? The description of the flowering time results needs to be clarified. From the text one can get the impression that the analysis of flowering time in the field confounded variation in flowering time (within a given flowering season) and year of first flowering, whereas correlations presented in the supplement suggest that flowering times recorded in different years (i.e., of plants of different age(?)) were analysed separately, as would be appropriate. Please, clarify! Moreover, it is not clear what “this trait” on the penultimate line of p. 10 refers to.

Response to 1.18)

For clarification, we have added the following “In the field, we found the flowering time characteristics to depend greatly on the year and the planting date, and we analyzed data from each planting year separately.”. In addition, we have added a new figure to better illustrate the flowering time traits. See point 1.19 for a comparison to the flowering of natural populations in the region.

1.19) p. 10, third paragraph. “Many of the northern accessions failed to flower in the planting year”. Does this imply that flowering the first year was maladaptive? This also suggest that the GWA of flowering time in the first year was based on data for fewer accessions compared to GWA of flowering time in subsequent years? May this have affected the results? Please, clarify.

Response to 1.19)

The flowering time pattern for Lotus is complex with multiple rounds of flowering per year. We agree that this was not well described in the previous version of the manuscript. To remedy this, we have included an illustration of flowering time in the new Figure 3B and stated in the legend: “Flowering time 2014 was quantified as the days to first flowering since planting. If the plants did not flower in the planting year, the days from planting in 2014 until flowering in 2015 were counted.”

It is an interesting question whether flowering in 2014 was maladaptive, since many of the northern accessions did not flower at a northern site. We think the lack of flowering in 2014 for northern accessions reflects that they are aware when winter is coming and will not flower outside the right window of opportunity. Because of the late planting date in 2014, no plants flowered before August, which is 2-3 months later than the natural populations in the region. In 2015, when the plants were well-established, they all flowered nearly synchronously in May at the same time as the natural populations. In 2015, the adaptation of the northern accessions was reflected by their reduced duration of the second flowering time period, again reflecting the anticipation of winter. See the new Figure 3B. We therefore think that the two traits Flowering time 2014 and Duration of the 2nd flowering time period both reflect winter anticipation, which is likely to be an adaptive trait in northern Japan. We have added these reflections to the discussion section of the manuscript.

1.20) p. 19. The authors should provide more details regarding experimental design and analysis: Seeds were planted in the field in four consecutive years. Was the analysis of flowering time in different years conducted separately for plants of different age? From the description it seems that, in at least some years, plants of different age were grown together. What were sample sizes per accession in different years? How was flowering duration defined?

Response to 1.20)

To clarify, we have stated that: “We grew the accessions for three consecutive years at the field site, and each year we replanted to replace the plants that did not survive winter. When quantifying phenotypic traits, only plants of the same age were compared.”.

In addition, we have illustrated how flowering time duration was defined in the new Figure 3B “The duration of the second flowering time period was quantified as the number weeks flowers were observed after senescence of the first set of pods. The pods senesced in July 2015 for most accessions.” The scoring details for all traits are also listed in Supplemental file 1 under the tab Phenotype descriptions.

1.21) Graphs such as Fig. 3C are nice because they give a broad overview of geographical variation, but they should be complemented with tables giving means, SD, and sample sizes for individual accessions as supplemental material.

Response to 1.21)

We have added the requested data to Supplemental file 1 under the tab “Phenotype data detailed”.

1.22) p. 11. What was flowering time in the field experiment in 2014 compared to that of natural populations in the area? The authors suggest that the flowering time in the field experiment that year was mainly a consequence of the late planting. This suggests that this trait may represent ability to rapidly reach reproductive maturity rather than flowering time of reproductive individuals. This should be considered when interpreting the results.

Response to 1.22)

Thank you for pointing this out. We have now thought more carefully about the interpretation of the flowering time data. Please see Response to 1.19) for our conclusions.

1.23) p. 11, final paragraph. Aren't high F_{st} expected for top SNP associated with traits that differ between “subpopulations”? Can this be considered an “independent test of local adaptation” when GWA and F_{st} analyses are based on the same set of accessions (see also Discussion first paragraph, “the power of this combination”)?

Response to 1.23)

They are independent analyses in the sense that the Fst analysis does not include phenotype information, whereas the GWA analysis does not use the results of the population structure analysis as input. The GWA analysis corrects for population structure using a kinship matrix in order to eliminate associations that can be explained by population structure alone. The basic premise of our study has been that dependence is then only created by local adaptation since this is expected both to be driven by phenotypic differences and cause increased population differentiation.

However, as also noted by other reviewers, population structure correction might not be perfect, and in the previous version of the manuscript, we were lacking a reliable measure of how frequently our observed Fst distribution skews could be generated by chance. We have therefore now carried out a permutation analysis, randomly re-assigning phenotypes to accessions, for all traits. Based on these permutation analyses, we calculate false discovery rates for the Fst distribution skews and Fst/p-value correlations. Please also see Response to 3.2).

1.24) p. 12, second paragraph. For the same reason, because pop1 and pop2 did not differ markedly in overwintering survival, it shouldn't be surprising that Fst values for overwintering SNP in the comparison between pop1 and pop2 were not high. Or am I missing something here?

Response to 1.24)

That is correct. It would, however, be surprising if the Fst values were high in the pop1/pop2 comparison, which would suggest that population structure effects were generating false positive GWAS signals. This is why we tested and why we were reassured that no skew towards high Fst values were observed.

1.25) p. 12, second paragraph, final sentence, "this environmental variable was likely critical...". Or some other environmental factor correlated with minimum temperature!

Response to 1.25)

Yes, that is a good point. We have added the statement as suggested.

1.26) p. 15, second paragraph, second sentence. Move "for winter 2014" to after "GWA signals"?

Response to 1.26)

The suggested change was made.

1.27) p. 19, first paragraph. How are the 136 accessions distributed across sampling sites (“local populations”)? Were all from distinct sites or did some local populations contribute more than one accession?

Response to 1.27)

Yes, some local populations did contribute more than one accession. We have now clearly indicated in Supplemental file 1 for which accessions this was the case. When the field experiments did not include the full set of accessions, we selected against inclusion of multiple accessions from the same local population.

1.28) p. 20, first paragraph. The authors state that 136 accessions were planted in 2017. How many accessions were included in the field experiments established in each of the preceding years?

Response to 1.28)

We have now listed the number of accessions scored for each phenotype in the Phenotype descriptions tab of Supplemental file 1.

1.29) p. 20, second paragraph. Does this species need vernalisation for flowering? Does vernalisation requirement vary among accessions as is the case in other species? Did the greenhouse plants receive any stratification/vernalisation treatment as would most likely plants in natural populations?

Response to 1.29)

To the best of our knowledge, none of the Lotus accessions require vernalisation for flowering. The greenhouse plants did not receive any vernalization treatment and all flowered relatively quickly. We have now stated this in the manuscript.

Reviewer #2 (Remarks to the Author):

This manuscript studied the local adaptation of Lotus japonicus in Japan. Firstly, authors resequenced 136 accessions, and performed detailed analyses of population genetics. Secondly, authors phenotyped these accessions in field and also in the growth house. Lastly, they made GWAS analysis based on different traits that could be correlated with adaptation, and looked at the overlap between highly differentiated regions (Fst) and GWAS signals. In general, this is a very good study about local adaptation. However, I have some concerns and comments about the manuscript below.

Major:

2.1) In Figure 1 and 2, it would be better to reorganize the two figures, since both shows the results of population structure and differentiation, but not well-organized. We studied the population structure of the populations firstly (mostly Figure 2C), then based on results we could divide samples into different populations.

Response to 2.1)

We have actually had the figures organized in the way you suggest in previous versions of the manuscript, but found that the storyline was easier to follow in the current configuration. Noting that the other reviewers have not objected to the organisation of these figures, we have opted to retain current layout.

2.2) PSMC analysis will be robust reveal more early evolution history than MSMC, it is better to add the MSMC analysis as comparison to reveal the demographical history of Lotus japonicus. This is highly rewarding given Lotus japonicus distributed to Japan very recently.

Response to 2.2)

It is a good point that MSMC is more accurate than PSMC for recent time periods, but this increased power comes from simultaneous analysis of multiple individuals. Our analysis is based on pseudodiploid pairs, and we can therefore not meaningfully increase the number of individuals to take advantage of the MSMC approach. It is true that when applied to human data, for example, the PSMC estimation is inaccurate for the past 10,000 years, where MSMC provides more reliable estimates. However, human generation time is roughly 30 years. The generation time of Lotus is roughly one year, meaning the PSMC analysis in Lotus should be fairly accurate up to about $10,000 / 30 = 333$ years ago, and we only make conclusions in the time interval between 2000 and 15000 years. We have now mentioned this effect of the generation time in the manuscript.

2.3) For the analyses of *Fst* and GWAS, the cut-offs are not clear. What is the rationale to use a cut-off of p -value $\leq 5e-5$ for GWAS? Usually people calculate the cut-off as $0.01/\text{total SNPs}$. For *Fst* analysis, authors mentioned $Fst > 0.65$, but what is the rationale behind this value as a cut-off. Given both of these analyses are fundamental results for the main conclusion of this study, it is important to carefully describe these methods.

Response to 2.3)

It is a good point that our cutoffs were not well-defined or explained in the previous version of the manuscript. We have now indicated the Bonferroni threshold in all GWA Manhattan plots and use the top 200 SNPs from all GWA and permutation runs in order to get comparable false discovery estimates for the *Fst* distribution skews.

2.4) Although this study did not perform the functional validation like forward genetics, it is helpful to discuss the candidate genes in more detail, based on the function of orthologous gene in *Arabidopsis* or other well-studied plants.

Response to 2.4)

We have added more thorough descriptions of the putative candidate gene functions and indicated where no functional information is available.

2.5) In the results section the third paragraph, “tight clustering ...”, it would be better to make the claim based on π value at the genome level, actually authors also mentioned π value in Figure 2E.

Response to 2.5)

We see your point, but do not think that we make any claims based on the tight clustering in the PCA plot. We only state that it suggests lower genetic diversity and later go on to test that explicitly.

Minor:

2.6) In Figure 1, it is better to explain what are the meaning of the different colors.

Response to 2.6)

We have added the following to the figure legend: “A-B) Accessions are colored according to their grouping in the PCA plot.”

2.7) In Figure 1D-E, samples MG110 and MG111 were not marked on the figure, it is hard to recognize and follow here.

Response to 2.7)

We have highlighted the positions of MG110 and MG111 in Figure 1D-E.

2.8) The order of the citation of Figures need to be updated, Figure 4 shows up in the main text early than Figure 3, authors need proofread and correct these kinds of problems.

Response to 2.8)

Is it possible that you might have read Supplementary figure 4 as Figure 4? In any case, we have included a new figure, double checked that all figure references are correct, and that figures are correctly ordered.

2.9) It is better to mark all the genes mentioned in the main text that are overlapped between Fst and GWAS on the figure 5.

Response to 2.9)

That is a good idea. We have indicated the positions of all genes discussed in the text on Figure 5 (now Figure 6).

2.10) It takes much more effort to understand the figure 4, it is better to explain the figure in the figure legends in detail. Also in Figure 5, the meaning of dots in different colors should be explained clearly.

Response to 2.10)

We have explained the figures in greater detail in the figure legends as suggested.

Reviewer #3 (Remarks to the Author):

Shah et al. study the process of the historical colonization and range expansion in Lotus japonicus in Japan using a combination of population genomics and common garden studies. The project is motivated by an interest in the genes impacting local adaptation. Can they be discovered and their evolutionary history explored? The project relies on the collection of a representative accession panel and shallow resequencing of this material, along with studies of population structure and trait associations. Together, these data build a picture of the timing and order of range expansion and suggest a number of potential environmental factors and plant traits impacting adaptation. Overall, this is a nice contribution within the context of the study system.

*The project centers on using F_{st} measures of differentiation and coalescent analyses to understand population structure and genetic ancestry of population. Overall, the analyses identify several strongly differentiated groups (median $F_{st} \sim 0.25$) and infer that Kyushu Island is the center of diversity and origin of colonization of *L. japonicus* in Japan. The manuscript discusses some divergence times and builds a plausible demographic history of these populations. The manuscript then focuses on discussing genes with extreme F_{st} statistics as likely candidates underlying divergent selection and leading to adaptation between subdivided populations. This is a relatively common approach that can be informative for generating hypothesis. However, it should be acknowledged that there will always be outliers and the tail of the neutral F_{st} distribution can be remarkably long (high F_{st} by chance alone; see various papers by Mike Whitlock for detailed discussion).*

3.1) The manuscript provides no real guidance as to what F_{st} values might be expected under purely neutral processes.

Response to 3.1)

Modeling F_{st} distributions under neutral processes would be an interesting study in itself, but here we focus on the F_{st} distributions of groups of SNPs associated with phenotypic traits. We think this approach is a good alternative to using modeling of F_{st} distributions to determine a significance cutoff that would distinguish between selective and neutral processes, which we think would be quite challenging. Actually, in the absence of any phenotypic information we believe that it would be hard to argue that the F_{st} distribution we report could not have been generated by neutral processes alone.

The novel aspect of this manuscript is that the accession panel was planted in common garden in the field and that a series of phenotypes thought to be important in adaptation were screened. As such, the group

can ask whether loci with effects on candidate traits are also highly differentiated. These data are nicely presented in Figure 4. The inference is that the joining of these data lead greater credence in identifying candidate genes and their role in local adaptation and subsequent differentiation. I think this is so, but only up to a point.

One major limitation in this endeavor is the potential confounding effects of population structure. GWAS is premised on the assumption that associations between marker and traits are caused by physical linkage generating linkage disequilibrium (LD). Unfortunately, many other evolutionary processes can also impact allele frequencies and LD causing spurious associations between markers and traits. The best scenario for GWAS is therefore in a randomly mating/panmictic population as here the major driver of LD is physical distance. In this case, observed correlations between markers/traits are likely to lead you to causal genes. However, most systems have some degree of population structure. Depending on the severity of population structure, some false positive associations are likely to occur. The problem is sorting out which hits are false positives versus real causal genetic signal. The problem can be really severe if the traits of interest are differentiated in the same direction as global population structure (as is the case here). In this case, any polymorphism that shows allele frequency differences by population structure will show an association with the similarly differentiated trait.

3.2) GWAS analyses often include population structure as a covariate in the model. However, we are never able to fully “control” for population structure, and in some cases including it in the analyses can actually reduce the ability to see real signal. The authors I’m sure are well aware of these limitations.

However, I felt like much of the text is written as if these are minor problems that are adequately addressed by including kinship in the analyses. The same problem holds for looking at the joint pattern of Fst outlier and GWAS signal. I think the manuscript needs to carefully address this limitation.

Response to 3.2)

Thank you for laying out these valid concerns so clearly. Since we observed different sets of highly associated SNPs across the traits investigated, we were confident that the kinship-based population structure correction was working reasonably well. If population structure were not properly corrected for, we might have seen the same SNPs showing up as associated with all the traits correlated with population structure, which was not the case. However, in the previous version of the manuscript, we did not present data that allowed us to evaluate the false discovery rates of the observed Fst skews.

We have now carried out a permutation analysis in order to estimate false discovery rates, as you suggest in point 3.8). The simulation strategy accounted for population structure by sampling phenotypes with a given covariance, where the covariance matrix was set to the estimated covariance matrix ($\sigma_g^2 K + \sigma_e^2 I$) obtained from the linear mixed model fit on the observed trait. This sampling strategy ensures

that the sampled phenotypes display similar levels of correlations with population structure as the observed phenotype, thus reducing potential of bias due to population structure. The validity of this approach has been tested previously (<https://www.ncbi.nlm.nih.gov/pmc/articles/PMC4634896/>).

We found low *fdr* values for the central traits discussed, and we think that the addition of the permutation analysis has greatly strengthened the manuscript, since it presents the reader with a tangible measure of the significance of the findings.

To underline the concerns with respect to the influence of population structure, we have now added the following to the Results section: “For the traits that were strongly correlated with both geographical origin and population membership (Supplemental figure 5), we were concerned that population structure might cause *Fst* distribution skews unrelated to the phenotype examined, despite the correction for population structure in the GWA analysis. To investigate if this was the case, we carried out GWA analysis using permuted phenotype data for each trait. The permutation was carried out such that plant genotype and phenotype were uncoupled while ensuring that the permuted phenotypes displayed similar levels of correlation with population structure as the observed phenotypes. Based on the results of 1000 permutations, we then calculated false discovery rates for the observed *Fst* skews for each trait.”

And we have stated in the Discussion: “Because of the complex effects of population structure on GWA results, permutation analysis, where the correlation between phenotype data and population structure was maintained, was important in generating trait-specific empirical data that allowed calculation of significance measures to support our conclusions. ”

I have a number of other specific comments:

3.3) The manuscript mentions a number of divergence/cross coalescence dates but never gives a sense for the uncertainty in these estimates. Is it possible to put confidence intervals or bounds to these estimates?

Response to 3.3)

PSMC analysis does not easily provide standard errors without imposing further assumptions. Furthermore, our point is not to make strong claims on the cessation of gene flow for a given comparison of accessions but to look for patterns in the large pairwise matrix of cessation of gene flow. The two specific examples of comparison for single accessions are just provided to give a visual idea of the signal available in the data. We believe that the patterns in the matrix are relatively clear. We do not want to put too much emphasis on specific times since they depend on the mutation rate used for calibration, which also have significant uncertainty. We do suggest that the ranges of separations are consistent with colonisation after the end of the last glaciation.

3.4) *The manuscript assumes the reader knows a bit about Lotus natural history. It might be good to include a paragraph (at least in a supplement) about the mating system, life history, and basic ecology of Lotus.*

Response to 3.4)

We have now mentioned in the introduction that Lotus is a self-compatible, diploid wild perennial legume found in diverse natural habitats. In addition, we have added a new Figure 3 that illustrates the annual life cycle of Lotus as it withers away in fall and regrows in spring. This figure also summarises the flowering behaviour for the different accessions. We think that this additional figure nicely describes and clarifies the central phenotypes under investigation.

3.5) *The Emma model used for association mapping is pretty conventional – however, there are other approaches that could be considered (Q + K, PCAs). Given the strong population structure in the system it would be good to justify the basic Emma kinship model is best. Perhaps some comparisons could be included in a supplement.*

Response to 3.5)

We chose instead to focus on the widely used and accepted EMMA kinship model and carry out permutation testing to investigate the limitations. See also Response to 3.2)

3.6) *As far as I can tell, none of the statistical tests are controlled for multiple testing. Only nominal p-values are provided. The authors should at least address this concern and provide their philosophy for interpreting genomewide p-values.*

Response to 3.6)

We have now indicated the Bonferroni 0.05 multiple testing correction line on the GWA Manhattan plots. In addition, we now use the top 200 SNPs from all GWA and permutation runs in order to get comparable FDR estimates for the Fst distribution skews.

3.7) *One way to get a feel for the degree of population confounding is simply to ask how well population structure alone can predict phenotypes. This can be accomplished by predicting phenotype from population membership coefficients, for example. The best case scenario would be no relationship...*

Response to 3.7)

The trait correlations with population structure are listed in Supplemental figure 5. The variance explained equals the squared correlation coefficient, that is $0.79^2 = 62\%$ for overwintering 2014 and $0.64^2 = 40\%$

for flowering time 2014. So, population membership is by no means a perfect predictor. On the other hand, if there were no correlation, the traits would be unlikely to be associated with local adaptation.

3.8) The manuscript mentions “enrichments” in several contexts – however, p-values are never presented in the text for those enrichments. Can randomization tests be used to explore the overlap of GWAS LOD and Fst outlier statistics? It might be nice to see how often these associations can arise by chance.

Response to 3.8)

Thank you for suggesting this approach. We have now carried out the permutation analysis, which we think has greatly strengthened manuscript. See point 3.2).

3.9) Phenotypes were collected from common gardens over multiple years. However, it seems each year is analyzed separately. It may be that power could be increased if these data are joined in a mixed model. Mortality from the first year is interpreted as freezing tolerance and overwintering survivorship. However, it could also be related to transplant shock and general vigor. Is there any way to disentangle these alternatives?

Response to 3.9)

Yes, each year was analysed separately. We also tried analysing combined data from multiple years, but it did not increase signal strength and for simplicity, we have not included those analyses in the manuscript. Probably due to the late planting date and harsh winter conditions in 2014, we saw the largest level of differentiation between accessions and strongest GWA signals that year.

We have now clarified that only well-established plants were evaluated for winter survival, allowing us to measure winter survival rather than general vigor or transplant shock effects. To better illustrate the winter survival phenotype, we have added a new Figure 3 showing field images of winter hardy and -sensitive accessions across a growth season.

3.10) My understanding is some of the plant material died across successive common garden years. One of the ideas is that this loss represents selection from freezing susceptibility (it's not random). As such, doesn't this selection potentially cause problems in the GWAS analysis as it may be generating strong LD? The manuscript mentions some replanting.....but the details weren't clear to me.

Response to 3.10)

We only evaluated plants transplanted at the same time and have now clarified this in the manuscript. That is, all plants were evaluated for winter survival in spring 2015 (OW 2014). In spring 2016, only plants transplanted in 2015 were evaluated for winter survival, which is why there are fewer data points in later years, except for 2018, where the whole experiment was replanted in 2017.

3.11) The discussion section mentions evidence for “tradeoffs” I don’t think the manuscript currently provide any insight or evidence for tradeoffs and how they’d maintain variation.

Response to 3.11)

The mention of tradeoffs was removed.

3.12) The details of the experimental design of the common gardens were not clear. The methods section mentions planting 3 seedlings per pot and then each line being represented by two “spots”. So, is the pot the unit of replication?

Response to 3.12)

We have clarified the experimental design of the common garden experiment in the Methods section. It now reads: “Three individuals from the same accession were planted spaced least 30 cm apart directly in field soil. Subsequent sets of three individuals per accession were planted at 60 cm intervals. Each accession was planted in two sets of three, a total of six plants per experiment, in control and salt stressed fields.”

3.13) The salt stress experiment is not really mentioned or explored in the text – this seemed a bit odd.

Response to 3.13)

The salt stress experiment will be examined in detail elsewhere, but in the context of this study, it provided a number of useful control traits that did not appear to be associated with local adaptation.

Reviewers' comments:

Reviewer #1 (Remarks to the Author):

In this revised version, the authors have addressed a number of points received on the original submission. This has clarified several issues. I suggest there are a few things the authors should consider in a further revision.

Perhaps most importantly, I suggest the authors reconsider the analysis of flowering time data for the first set of plants put out in their field experiment. In the legend of Fig. 3, the authors state "Flowering time 2014 was quantified as the days to first flowering since planting. If the plants did not flower in the planting year, the days from planting in 2014 until flowering in 2015 were counted." This does not make sense to me because it confounds differences in flowering propensity in the first year (proportion of plants flowering in 2014) and differences in flowering time in the second year. I suggest the authors rather examine the two traits separately: first, proportion of plants flowering in the year of planting (2014), and second, flowering start in the second year. The sample size for flowering start in the second year will be reduced by the substantial winter mortality in some populations, but that is how it is.

Moreover, to understand which population had the highest overall fitness, it would be interesting to have data on fecundity in the first and in the following years. Was any attempt made to score fecundity of surviving plants?

Some additional comments:

P. 4, line 64. I do not agree with this and the following statement. Evidence of local adaptation is obtained if, in a reciprocal transplant, local genotypes have higher fitness than have non-local genotypes. A comparison of performance at "home" and "away" sites is not sufficient (see ref 1, Kawecki and Ebert 2004).

P. 4, line 79. I suggest the authors provide some more details regarding this argument.

p. 5, line 96. Change "eroded" to "weakened"?

p. 5, line 98. Add "by GWA" after "studied"? Certainly, other approaches have provided clear evidence of plant traits contributing to adaptive differentiation.

Fig. 2. "Pairwise divergence times and genetic distances". Please, provide units in which these were measured.

Fig. 3, legend. "The duration of the second flowering time period was quantified as the number weeks flowers were observed after senescence of the first set of pods." Why not as the number of weeks between start and end of flowering in that year, which would conform better to the usual definition of duration of flowering. It is easy to see that the two measures would differ if flowering start varies.

Fig. 4A. I suggest the authors reorder the results: sort means first by year and then by population rather than the other way around. The reader could then easily compare the survival of different "populations" in each of the four years.

p. 8, second paragraph. The results presented in Fig. 1 D and E are rather difficult to apprehend. Is there any easier way to illustrate the points made in this paragraph?

p. 8, line 160. It seems unlikely that a perennial herb has a generation time of 1 year. Do the authors have any demographic data for *L. japonicus* to support this assumption?

p. 10, line 222. Was there any effect of plant age or size on winter survival?

p. 17, line 388. I suggest that these results also illustrate the need to score phenotypes in an environment that is relevant for the populations under study (i.e., in an environment as similar as possible to that of the natural habitat).

p.17, line 393. The authors found the strongest signals of adaptive differentiation in a year when plants were forced to establish later than plants in natural populations in the area and therefore were exposed to a particularly short growing season (and a shorter growing season than natural populations in that year). The authors may want to discuss briefly this observation. Could it be that what appears as adaptive variation among populations has been shaped by selection in years with extreme weather conditions in northern populations (with a very short growing season), conditions not captured by the years of the experiment?

In response to a comment by reviewer 3, the authors state that plants of the same genotype were planted in groups of three. Was this aspect of the experimental design considered in the statistical analysis?

Reviewer #2 (Remarks to the Author):

This manuscript studied the local adaptation of *Lotus japonicas* in Japan. Based on resequenced 136 accessions, authors performed population genetics analyses. Furthermore, authors performed GWAS analysis using different traits that might be correlated with adaptation, and looked at the overlap between highly differentiated regions (F_{st}) and GWAS signals. As a conclusion, authors suggested that they provide evidence that these traits (overwintering and flowering time) were direct targets of selection by local adaptation during the colonization process and point to associated candidate genes. The revision largely addressed and clarified all my concerns, although authors prefer to keep the former organization of Figure 1 and Figure 2, instead to reorganize the figures to show the population structure related graphs first.

Reviewer #3 (Remarks to the Author):

An earlier reviewer noticed a serious limitation of the study. The criticism centers on the fact that the study does not use reciprocal common gardens. As such, it is impossible to strictly interpret the results as supporting local adaptation because the comparisons among genotypes are limited to only a single site. Local adaptation is defined by the presence of trade-offs and divergent selection that can only be studied as a contrast of the fitness at home versus away sites. The authors acknowledge this limitation in their "response to review" statement, and adjust a single sentence (line 230) to say that ".....this is consistent with pop3 local adaptation, although we cannot rule out that pop3 would outperform pop1 and pop2 in other environments". This is a subtle distinction but an important caveat with implications for confidence in the narrative developed about colonization and adaptation. Unfortunately, the bulk of the manuscript (including the title) is still strongly worded with an interpretation of pop3 local adaptation through overwintering tolerance relative to pop1 and pop2.

Earlier reviewer comments focused on confusion about the experimental design. The authors address many of these concerns in the revision. One issue is that the experimental protocols changed from year to year in relation to planting regimes (plantings occurred over a 6 months period across the study years). This makes it very difficult to compare results among years – what is the appropriate

protocol for studying phenotypic traits and natural selection in the system? How realistic are late field season plantings? The main GWA results that support adaptation are evident from only one of the study years (2014) – a year when plants were planted extremely late in the season. The authors suggest that early plantings “thwarted” GWA analyses and that the stressful aspects of planting late in 2014 revealed the signature of local adaptation. This may be so, but it is only one of several different interpretations. As such, too much of the discussion section focuses on this interpretation.

It’s nice that the revised manuscript better justifies the various thresholds used, including now presenting strict Bonferroni cutoffs for association. The inclusion of the permutation based tests for F_{st} /traits associations is a nice addition, and bolsters arguments made from the genomics data.

We wish to thank the reviewers again for their thorough evaluation of our work and for the detailed and constructive suggestions, which have been very helpful in guiding the revision of the manuscript.

Reviewer #1 (Remarks to the Author):

In this revised version, the authors have addressed a number of points received on the original submission. This has clarified several issues. I suggest there are a few things the authors should consider in a further revision.

1.1) Perhaps most importantly, I suggest the authors reconsider the analysis of flowering time data for the first set of plants put out in their field experiment. In the legend of Fig. 3, the authors state "Flowering time 2014 was quantified as the days to first flowering since planting. If the plants did not flower in the planting year, the days from planting in 2014 until flowering in 2015 were counted." This does not make sense to me because it confounds differences in flowering propensity in the first year (proportion of plants flowering in 2014) and differences in flowering time in the second year. I suggest the authors rather examine the two traits separately: first, proportion of plants flowering in the year of planting (2014), and second, flowering start in the second year. The sample size for flowering start in the second year will be reduced by the substantial winter mortality in some populations, but that is how it is.

Response to 1.1)

That is a good point. As suggested, we have now analysed flowering proportion in 2014 instead of flowering time 2014. There results were very similar to those of the previous analysis and did not result in any changes to the conclusions. The first scoring of the plants in 2015 took place a little too late to capture the variation in flowering time onset accurately, and we have therefore not analysed the onset of flowering time in 2015.

1.2) Moreover, to understand which population had the highest overall fitness, it would be interesting to have data on fecundity in the first and in the following years. Was any attempt made to score fecundity of surviving plants?

Response to 1.2)

This is difficult for *Lotus* as pods are produced and shatter continuously, making it difficult to get an accurate seed count. We did not attempt to do this.

Some additional comments:

1.3) P. 4, line 64. I do not agree with this and the following statement. Evidence of local adaptation is obtained if, in a reciprocal transplant, local genotypes have higher fitness than have non-local genotypes. A comparison of performance at “home” and “away” sites is not sufficient (see ref 1, Kawecki and Ebert 2004).

Response to 1.3)

We have rewritten the statement to make it more precise. It now reads: “Experimental evidence for local adaptation can be obtained by quantifying the fitness of individuals from two or more populations in common garden experiments at two or more locations. If population A outperforms population B when grown in a common garden in population A’s native environment, and the converse is true in population B’s native environment, populations A and B are locally adapted to their respective native environments.”

1.4) P. 4, line 79. I suggest the authors provide some more details regarding this argument.

Response to 1.4)

To clarify the argument, we have rephrased the paragraph so it now reads: “If a potentially adaptive signal detected using a phenotype-independent genome scan for population differentiation overlaps with a GWA signal derived from common garden fitness data, the argument is that these signals would constitute two independent lines of evidence supporting local adaptation, one based on genomic and the other on phenotypic differentiation.”

1.5) p. 5, line 96. Change “eroded” to “weakened”?

Response to 1.5)

The suggested change was made.

1.6) p. 5, line 98. Add “by GWA” after “studied”? Certainly, other approaches have provided clear evidence of plant traits contributing to adaptive differentiation.

Response to 1.6)

The suggested change was made.

1.7) Fig. 2. *“Pairwise divergence times and genetic distances”. Please, provide units in which these were measured.*

Response to 1.7)

The header for figure 2 was incorrect, thank you for noticing this. We have changed it to “Population structure and genetic diversity.” We have also provided a definition of genetic distance in Figure 1 and 2 legends.

1.8) Fig. 3, legend. *“The duration of the second flowering time period was quantified as the number weeks flowers were observed after senescence of the first set of pods.” Why not as the number of weeks between start and end of flowering in that year, which would conform better to the usual definition of duration of flowering. It is easy to see that the two measures would differ if flowering start varies.*

Response to 1.8)

Although it would make good sense, we were not able to change the definition of flowering time duration, since the plants were scored too late the first time in 2015 to capture the variation in flowering time onset accurately. This is why we used pod senescence, which was carefully determined, as a reference point for the flowering time duration trait. We have therefore not changed the analysis of flowering time duration.

1.9) Fig. 4A. *I suggest the authors reorder the results: sort means first by year and then by population rather than the other way around. The reader could then easily compare the survival of different “populations” in each of the four years.*

Response to 1.9)

We have changed the figure as suggested and agree that it makes it easier to read.

1.10) p. 8, second paragraph. *The results presented in Fig. 1 D and E are rather difficult to apprehend. Is there any easier way to illustrate the points made in this paragraph?*

Response to 1.10)

We have tried our best to illustrate the results and have not been able to come up with an improved illustration. Any specific suggestions for improvements are welcome.

1.11) p. 8, line 160. *It seems unlikely that a perennial herb has a generation time of 1 year. Do the authors have any demographic data for L. japonicus to support this assumption?*

Response to 1.11)

We do not have precise demographic data for *Lotus* to accurately estimate the generation time. To indicate this uncertainty, we have now indicated a possible generation time interval of one to two years.

1.12) p. 10, line 222. *Was there any effect of plant age or size on winter survival?*

Response to 1.12)

There was no apparent effect of size for the first year plants, but the 2nd year plants showed an increase in winter survival rates compared to the first year plants from the same accession (for those that did not already show survival rates of 100% for the first year plants).

1.13) p. 17, line 388. *I suggest that these results also illustrate the need to score phenotypes in an environment that is relevant for the populations under study (i.e., in an environment as similar as possible to that of the natural habitat).*

Response to 1.13)

We agree and have added that comment to the discussion.

1.14) p.17, line 393. *The authors found the strongest signals of adaptive differentiation in a year when plants were forced to establish later than plants in natural populations in the area and therefore were exposed to a particularly short growing season (and a shorter growing season than natural populations in that year). The authors may want to discuss briefly this observation. Could it be that what appears as adaptive variation among populations has been shaped by selection in years with extreme weather conditions in northern populations (with a very short growing season), conditions not captured by the years of the experiment?*

Response to 1.14)

Yes, that very accurately captures our thoughts. We have now attempted to phrase this more clearly in the discussion, mentioning that colonization of the north likely took place when the climate was significantly colder than it is today.

1.15) *In response to a comment by reviewer 3, the authors state that plants of the same genotype were planted in groups of three. Was this aspect of the experimental design considered in the statistical analysis?*

Response to 1.15)

We did not differentiate between the six plants but measured phenotypes for all individuals and used survival rates (winter survival) and simple averages (other phenotypes) for GWA analysis.

Reviewer #2 (Remarks to the Author):

1) This manuscript studied the local adaptation of Lotus japonicus in Japan. Based on resequenced 136 accessions, authors performed population genetics analyses. Furthermore, authors performed GWAS analysis using different traits that might be correlated with adaptation, and looked at the overlap between highly differentiated regions (F_{st}) and GWAS signals. As a conclusion, authors suggested that they provide evidence that these traits (overwintering and flowering time) were direct targets of selection by local adaptation during the colonization process and point to associated candidate genes.

The revision largely addressed and clarified all my concerns, although authors prefer to keep the former organization of Figure 1 and Figure 2, instead to reorganize the figures to show the population structure related graphs first.

Response to 2.1)

Thank you for the clear and positive assessment of our work.

Reviewer #3 (Remarks to the Author):

3.1) An earlier reviewer noticed a serious limitation of the study. The criticism centers on the fact that the study does not use reciprocal common gardens. As such, it is impossible to strictly interpret the results as supporting local adaptation because the comparisons among genotypes are limited to only a single site. Local adaptation is defined by the presence of trade-offs and divergent selection that can only be studied as a contrast of the fitness at home versus away sites. The authors acknowledge this limitation in their “response to review” statement, and adjust a single sentence (line 230) to say that “.....this is consistent with pop3 local adaptation, although we cannot rule out that pop3 would outperform pop1 and pop2 in other environments”. This is a subtle distinction but an important caveat with implications for confidence in the narrative developed about colonization and adaptation. Unfortunately, the bulk of the manuscript (including the title) is still strongly worded with an interpretation of pop3 local adaptation through overwintering tolerance relative to pop1 and pop2.

Response to 3.1)

We appreciate the concern that we did not provide formal proof of local adaptation in the previous version of the manuscript and see how this limited the strength of the conclusions drawn. Meanwhile, we found that colleagues at Miyazaki University, which is located in the south of Japan, grew and recorded survival data for eight of the accessions, including two from population 3. Since all eight accessions were also grown at the northern Tohoku site, we could compare fitness at the two locations. We found a clear tradeoff, with pop3 individuals performing significantly worse at the southern site than the non-pop3 individuals, providing support for pop3 is local adaptation. The reciprocal common garden data is presented as a new Figure 4. We think that the inclusion of this data has strengthened the manuscript and that the wording throughout, including the title, is now justified.

3.2) Earlier reviewer comments focused on confusion about the experimental design. The authors address many of these concerns in the revision. One issue is that the experimental protocols changed from year to year in relation to planting regimes (plantings occurred over a 6 months period across the study years). This makes it very difficult to compare results among years – what is the appropriate protocol for studying phenotypic traits and natural selection in the system? How realistic are late field season plantings? The main GWA results that support adaptation are evident from only one of the study years (2014) – a year when plants were planted extremely late in the season. The authors suggest that early plantings “thwarted” GWA analyses and that the stressful aspects of planting late in 2014 revealed the signature of local adaptation. This may be so, but it is only one of several different interpretations. As such, too much of the discussion section focuses on this interpretation.

Response to 3.2)

It is an interesting question what the appropriate protocol is for studying phenotypic traits and natural selection in the system. This was unknown to us at the point when the field experiments were initiated, which is why we investigated multiple different options. In hindsight, it makes sense to us that the growth conditions have to be sufficiently challenging to differentiate between adapted and non-adapted accessions in order to study adaptation and that these conditions do not necessarily correspond to those experienced by current natural populations. The reason could be that fixation of the winter hardiness alleles in pop3 accessions in northern Japan likely took place under harsh temperature conditions following the last glaciation period. The colder temperature would have pushed the germination date to later in the year and made low winter temperatures set in earlier, thus reducing the period of time *Lotus* seedlings had to establish before winter and imposing a more stringent selection for winter hardiness than what is currently affecting natural populations in the area. In line with the suggestions from reviewer 1 (see point 1.14), we have added these reflections to the discussion.

Since we have now also provided evidence for local adaptation (see point 3.2), we find it difficult to come up with likely alternative interpretations of the results. We are happy to discuss other interpretations that might have escaped us, in case there are specific suggestions.

3.3) It's nice that the revised manuscript better justifies the various thresholds used, including now presenting strict Bonferroni cutoffs for association. The inclusion of the permutation based tests for Fst/traits associations is a nice addition, and bolsters arguments made from the genomics data.

Response to 3.3)

Yes, including the permutation tests certainly increased our confidence in the results, and thank you again for the suggestion.

Reviewers' comments:

Reviewer #1 (Remarks to the Author):

In this revised version, the authors have clarified a number of points in response to comments received on the previous submission. However, a number of things are still unclear and should be addressed:

Line 168. In modelling the migration history of Lotus, the authors assume a generation time of "one to two years". The study species is a perennial herb, and populations are likely to include individuals that are markedly older than 1-2 years and to have overlapping generations. How strongly does the assumption of generation time affect the results? Would results differ markedly if the generation time was set to twice as long or longer?

Line 234. The authors should give more details regarding this analysis. First, they should state which independent variables were included in the "generalized linear model ANOVA". In response to a comment on the previous version, the authors state that GWA was based on means (I assume accession means), in which case the planting of accessions (two groups of three plants; line 533) is not an issue in the GWA. However, for testing phenotypic differences between accessions, the planting scheme should be considered in the statistical model (the three plants in a group cannot be considered independent observations). Second, in the legend to Fig. 4, the authors state that the boxplot shows winter survival "for non-admixed individuals". The authors should clarify two things: (a) Does each symbol represent the mean of an accession? (b) What were sample sizes when the analysis was confined to "non-admixed individuals" (cf. sample sizes given on p. 24)?

Line 539. The comparison with the results from the Miyazaki garden is problematic for several reasons and needs to be better described and motivated. First, as I understand from information in Supplementary Table 2, statistical tests are restricted to 5 of the 8 accessions in the Miyazaki garden. This should be motivated and described in the main text. In addition, the authors should indicate whether the survival of these accessions were typical of those of other "pop3 and non-pop3 accessions" grown at the northern site. Second, plants in Miyazaki were planted in 2017, but their survival were compared to survival at the northern site in three other years (data pooled across years), excluding the 2017 planting at the northern site. I think it would be preferable to compare results in Miyazaki to each of the planting years at the northern site. Third, this analysis should take into account the non-independence of plants in the same "triplet" in the garden and the non-independence of plants of the same accession. Apparently, the current analysis does not do that. Fourth, how does the planting date at Miyazaki compare to the phenology of natural populations in that region (cf. concerns regarding planting times at the northern garden)? Fifth, please clarify which accessions contribute to survival estimates presented in Fig. 4C -- only accessions included in the tests presented in Supplementary Table 2, or all "pop3" and all "non-pop3" accessions?

Finally, I still find the authors' use of the terms "local adaptation" and "adaptive traits" a bit fuzzy. The field experiment at the northern site detected a markedly higher survival and flowering propensity in local populations and interesting GWA signals in a year when plants were planted very late in the season compared to the situation in natural populations in that region, but not in the other years when the planting phenology was more in line with the phenology of natural populations. To me this suggests that the local populations do not show evidence of local adaptation during current environmental conditions. The results might be explained by intermittent strong selection under particularly cold seasons, in which case it could still be argued that results are consistent with populations being adapted to local conditions, but no results are presented to support such a hypothesis. Were winter conditions during the period of the experiment, (2014-2017) particularly warm compared to any long-term weather data available for this region? Alternatively, as suggested by the authors, current differentiation in overwintering ability may reflect adaptation to climatic conditions that prevailed previously, for example, during the period of colonization. In this case,

population differentiation in traits affecting overwintering ability may once have been driven by selection, although an adaptive advantage can no longer be detected when plants are exposed to "currently relevant conditions". There is thus no strong evidence that these traits are "adaptive" under current conditions. What is known about climatic variation in this area since the last glaciation? Has climate gone through both colder and warmer periods, and how does the timing of climatic changes compare to the inferred history of the study species in Japan?

Comments on details:

Line 60. Delete "by local adaptation". These words do not add any information here.

Line 88. The authors may here also want to cite Price et al. 2018 PNAS 115:5028–5033.

Line 127. Insert "likely" before "direct targets".

Line 264, "duration ... of flowering". The response by authors suggests that this variable should be named "end of flowering" rather than "duration of flowering" because start of flowering is unknown.

Line 404, "rather than for instance ...". I would suggest this depends on the question addressed.

Line 413, "The three traits..." Unclear what this means since I would argue that local adaptation to current conditions has not been demonstrated.

Line 443, "is acting". Change to "has acted".

Line 531. Were the positions of accessions in the common garden randomized? Please, state explicitly.

We wish to thank the reviewer for his/her patience and continued effort in helping us improve the quality of the manuscript. Throughout the review process, the comments have been constructive and have provided clear directions for revision, which has been much appreciated by the authors.

Reviewer #1 (Remarks to the Author):

In this revised version, the authors have clarified a number of points in response to comments received on the previous submission. However, a number of things are still unclear and should be addressed:

1) Line 168. In modelling the migration history of Lotus, the authors assume a generation time of “one to two years”. The study species is a perennial herb, and populations are likely to include individuals that are markedly older than 1-2 years and to have overlapping generations. How strongly does the assumption of generation time affect the results? Would results differ markedly if the generation time was set to twice as long or longer?

Response to 1:

In the modeling of the migration history, we use the mutation rate per year (... , assuming a 159 mutation rate of 6.5×10^{-9} per year.), so the generation time does not affect the migration modeling. We have now stated this explicitly in the manuscript: “Note that the generation time does not affect the time estimates for cessation of gene flow, since these are based on the mutation rate per year and mutations accumulate continuously throughout the lifetime of the plant.”

2) Line 234. The authors should give more details regarding this analysis. First, they should state which independent variables were included in the “generalized linear model ANOVA”. In response to a comment on the previous version, the authors state that GWA was based on means (I assume accession means), in which case the planting of accessions (two groups of three plants; line 533) is not an issue in the GWA, However, for testing phenotypic differences between accessions, the planting scheme should be considered in the statistical model (the three plants in a group cannot be considered independent observations).

Response to 2:

To take into account the blocked experimental design, we have now assigned a unique ID to each block of three plants and re-analysed the data using this block ID as a random effect. We have described this in a new methods section “Statistical analysis of winter survival”. The blocks do explain some of the variation in winter survival, but the population differences remain highly significant both overall and for individual years. We have also provided the source data for this analysis, listing the field position, block ID and survival status for all tested individuals (“Field placement” tab in Supplemental file 1).

3) Second, in the legend to Fig. 4, the authors state that the boxplot shows winter survival “for non-admixed individuals”. The authors should clarify two things: (a) Does each symbol represent the mean of an accession? (b) What were sample sizes when the analysis was confined to “non-admixed individuals” (cf. sample sizes given on p. 24)?

Response to 3)

We have clarified in the manuscript that each symbol represents an accession mean in figure 4A and indicated the samples sizes, when the admixed accessions were excluded.

4) Line 539. *The comparison with the results from the Miyazaki garden is problematic for several reasons and needs to be better described and motivated.*

4.1) *First, as I understand from information in Supplementary Table 2, statistical tests are restricted to 5 of the 8 accessions in the Miyazaki garden. This should be motivated and described in the main text. In addition, the authors should indicate whether the survival of these accessions were typical of those of other “pop3 and non-pop3 accessions” grown at the northern site.*

Response to 4.1)

We have now indicated in the text that we compared survival of the accessions with full (MG030 and MG007) to those with no (MG008, MG066 and MG020) population 3 membership, excluding the accessions that show population 3 admixture (Gifu, MG076, and MG005) from the analysis. The mean survival of the pop3 and non-pop3 individuals from the above-mentioned groups were similar to those of all pop3 and non-pop3 individuals grown at the northern site. We have stated this in the manuscript and added the mean survival rates of all grown accessions to Supplementary Table 2 for reference.

4.2) *Second, plants in Miyazaki were planted in 2017, but their survival were compared to survival at the northern site in three other years (data pooled across years), excluding the 2017 planting at the northern site. I think it would be preferable to compare results in Miyazaki to each of the planting years at the northern site.*

Response to 4.2)

In addition to the analysis of the pooled Tohoku data, we have now included the results of analysis for each individual year in Supplementary Table 2. All four comparisons were significant for all three years, except for pop3 performance at Tohoku versus Miyazaki in 2016.

4.3) *Third, this analysis should take into account the non-independence of plants in the same “triplet” in the garden and the non-independence of plants of the same accession. Apparently, the current analysis does not do that.*

Response to 4.3)

We have now included the experimental blocks in the statistical analysis, see responses to 2) and 4.2). The conclusions remain the same.

4.4) Fourth, how does the planting date at Miyazaki compare to the phenology of natural populations in that region (cf. concerns regarding planting times at the northern garden)?

Response to 4.4)

This is difficult to evaluate as plants stay green and produce seeds all year round in the south, making it challenging to distinguish specific cycles and the timing of plant establishment.

4.5) Fifth, please clarify which accessions contribute to survival estimates presented in Fig. 4C - only accessions included in the tests presented in Supplementary Table 2, or all "pop3" and all "non-pop3" accessions?

Response to 4.5)

We have clarified in the figure legend that the graph is based on data from five accessions grown at both sites (MG020, MG066, MG008, MG007 and MG030).

5) Finally, I still find the authors' use of the terms "local adaptation" and "adaptive traits" a bit fuzzy. The field experiment at the northern site detected a markedly higher survival and flowering propensity in local populations and interesting GWA signals in a year when plants were planted very late in the season compared to the situation in natural populations in that region, but not in the other years when the planting phenology was more in line with the phenology of natural populations. To me this suggests that the local populations do not show evidence of local adaptation during current environmental conditions. The results might be explained by intermittent strong selection under particularly cold seasons, in which case it could still be argued that results are consistent with populations being adapted to local conditions, but no results are presented to support such a hypothesis. Were winter conditions during the period of the experiment, (2014-2017) particularly warm compared to any long-term weather data available for this region? Alternatively, as suggested by the authors, current differentiation in overwintering ability may reflect adaptation to climatic conditions that prevailed previously, for example, during the period of colonization. In this case, population differentiation in traits affecting overwintering ability may once have been driven by selection, although an adaptive advantage can no longer be detected when plants are exposed to "currently relevant conditions". There is thus no strong evidence that these traits are "adaptive" under current conditions. What is known about climatic variation in this area since the last glaciation? Has climate gone through both colder and warmer periods, and how does the timing of climatic changes compare to the inferred history of the study species in Japan?

Response to 5)

We agree that this is an important discussion point in the manuscript, and we have now clarified the limitations of our conclusions with respect to local adaptation, considering the effect of the planting date and the current climatic conditions, by stating the following in the Discussion: "In 2017, when an earlier planting date was used that more closely approximated the phenology of natural populations, nearly all accessions survived winter. Winter survival, as quantified in 2014, might therefore not be an adaptive trait under current conditions at Tohoku, although it is possible that other factors such as pedoclimate, plant density and competition with other plant species could challenge plants in natural populations in a fashion that would reveal inter-accession differences in a similar fashion as the late planting date did in our field experiments". We think this is a fair statement, as we cannot fully match the complex environment experienced by natural populations in a transplantation experiment.

In addition, as indicated in the reviewer comment, we have already outlined the alternative possibility that the winter hardiness alleles were fixed during post-glacial colonization, when temperatures were lower than today and we have now also mentioned that traversal of the central mountain range may have played a role. The spatiotemporal resolution of the temperature record and inferred colonization route is not sufficiently high to allow a more detailed investigation.

Comments on details:

6) *Line 60. Delete “by local adaptation”. These words do not add any information here.*

“by local adaptation” was deleted as suggested.

7) *Line 88. The authors may here also want to cite Price et al. 2018 PNAS 115:5028–5033.*

The suggested reference was added.

8) *Line 127. Insert “likely” before “direct targets”.*

“Likely” was inserted as suggested.

9) *Line 264, “duration ... of flowering”. The response by authors suggests that this variable should be named “end of flowering” rather than “duration of flowering” because start of flowering is unknown.*

We have changed this throughout as suggested.

10) *Line 404, “rather than for instance ...”. I would suggest this depends on the question addressed.*

Yes, we agree. The text “, rather than for instance focusing on a single trait phenotyped across many individuals and subpopulations.” was deleted.

11) *Line 413, “The three traits...” Unclear what this means since I would argue that local adaptation to current conditions has not been demonstrated.*

We have rephrased this paragraph so it now reads: “Three of the traits that showed the most pronounced Fst skews, Overwintering 2014, Flowering proportion 2014 and end of flowering 2015 (Figure 6B), ...”

12) Line 443, “is acting”. Change to “has acted”.

The suggested change was made.

13) Line 531. *Were the positions of accessions in the common garden randomized? Please, state explicitly.*

Supplemental file 1 now contains the full details of the winter survival field experiment, including the row and column position of each of the accessions in the field. The accessions were planted in three different groups and ordered within each of these groups according to their accession ID. Although the accession ID is not related to geographical location, but to the time of collection, it was possible that the field setup could have influenced the phenotype. We tested that by evaluating the effects of the row and column positions, and found that they explained little of the phenotypic variation.

We have now stated the following in the main text: “In the field, the accessions were planted in groups of three in a row and column grid. Field row and column placement only had a minor effect on the survival phenotype (<3%), whereas the planting groups explained 13% and the population membership 33% of the variation.”. We have described the analysis in the “Statistical analysis of winter survival” section of the Methods.

We have also described the field setup in more detail in the “Phenotyping” section of the methods, stating that “The accessions were subdivided into three groups, each covering a wide range of geographic origins. Within each of these groups, the accessions were planted ordered by their accession ID in the column direction in the field.”.

Reviewers' comments:

Reviewer #1 (Remarks to the Author):

The authors have responded to comments received on a previous version in an overall clear and satisfactory way, and as a result the presentation has improved considerably.

One important comment on the previous version regarded the statistical model used to analyse effects of "population" (pop3 vs. non-pop3) and "location" on survival of five accessions in experiments conducted at two field sites. The authors have now revised and clarified the model used (line 574), but one major aspect remains: I cannot see that the factor "accession" is considered (two accession representing "pop3" and three accessions representing "non-pop3" were included in the experiment). Because plants of a given accession cannot be considered independent observations of "population", this factor should be included in the model (nested within "population"). Moreover, I assume the population x location interaction was included in the model? What was the P-value associated with this interaction (should be given in the Results)?

Because the number of accessions per population is so low, an alternative way of analyzing the data would be to replace the factor "population" with accession and the population x location interaction with an accession x location interaction, and then use contrasts to ask whether the two "pop3"-accessions outperform the three "non-pop3"-accessions at the northern site, and whether the opposite is true at the southern site.

Conclusions should reflect the fact that a limited number of accessions of each "population" were tested.

Some additional comments on presentation:

Line 103. Not clear why a weak link between geographic and genetic distance would complicate identification of local adaptation. On the contrary it could be argued that a correlation between geographic distance and measures of differences in environmental conditions complicates the detection of signals of selection (cf. necessary control of population structure).

Line 257, "but worse...and the converse was true for non-pop3 individuals". Change to "while the converse was true at the southern Miyazaki site". Because your focus is on detecting evidence of local adaptation, the critical comparison is that between local and nonlocal genotypes at each of the two sites ("pop3 vs. non-pop3 survival at each of the sites"). Revise the following sentence reporting pairwise tests accordingly.

Lines 260-263. Similarly, the description of performance of the different accessions at the two sites should focus on how local and nonlocal genotypes performed at each of the two sites (not on how nonlocal genotypes performed at the two sites).

Line 267, "to maximize the chances of identifying phenotypic variation of adaptive relevance...". Not clear what the documentation of flowering time in the greenhouse adds in this regard. Please, explain or remove.

Line 268, "both a field and a greenhouse flowering time experiment". Indicate where the field experiment was conducted. The previous paragraph discusses field data obtained at two different sites.

Line 312. This sentence is difficult to understand. Consider possible rewordings.

I think several of the section titles push conclusions beyond what can be stated based on the results presented and that they therefore should be moderated:

"Population 3 is locally adapted to a cold climate" – I suggest change to something like "Population 3 is adapted to a cold climate" or "Population 3 has higher tolerance to cold"

"Selection acted on winter survival during pop3 local adaptation" – Data are consistent with this, but do not prove it. I would suggest that a more appropriate title would be "Winter survival and pop3 differentiation"

"Flowering and seed traits influenced pop3 local adaptation" – Change to something like "pop3 has diverged in flowering phenology and seed traits"

Line 382, "in a very regular fashion". Unclear what this means.

Line 387. Insert "evidence of" before "local adaptation".

Line 391, "nor did they account..." This is known since long and has been the topic of lots of papers before.

Line 407, "adapted and non-adapted accessions". Change to "local and nonlocal accessions".

Line 438. Insert "during plant development" after "set in earlier".

Line 438, "reduced the period of time Lotus seedlings had to establish before winter". Change to "reduced the time available for Lotus seedlings to establish before winter".

Line 576, "each genotype was grown in multiple blocks of three individuals". Please, reword to state explicitly that the three plants in a block represented three different genotypes (if that was the case).

Point-by-point response to reviewer comments

We would again like to express our appreciation of the reviewer's patience and attention to detail during the multiple rounds of reviews. Each iteration has brought significant improvements to the presentation and accuracy of the study, all of which are highly valued.

Reviewer #1 (Remarks to the Author):

The authors have responded to comments received on a previous version in an overall clear and satisfactory way, and as a result the presentation has improved considerably.

1) One important comment on the previous version regarded the statistical model used to analyse effects of "population" (pop3 vs. non-pop3) and "location" on survival of five accessions in experiments conducted at two field sites. The authors have now revised and clarified the model used (line 574), but one major aspect remains: I cannot see that the factor "accession" is considered (two accession representing "pop3" and three accessions representing "non-pop3" were included in the experiment). Because plants of a given accession cannot be considered independent observations of "population", this factor should be included in the model (nested within "population"). Moreover, I assume the population x location interaction was included in the model? What was the P-value associated with this interaction (should be given in the Results)?

Because the number of accessions per population is so low, an alternative way of analyzing the data would be to replace the factor "population" with accession and the population x location interaction with an accession x location interaction, and then use contrasts to ask whether the two "pop3"-accessions outperform the three "non-pop3"-accessions at the northern site, and whether the opposite is true at the southern site.

Conclusions should reflect the fact that a limited number of accessions of each "population" were tested

Response to 1)

It is a good point that the experimental data has a nested structure, which should be fully taken into account in the statistical analysis. The blocks in the previous model captured most of the accession effect, but the accessions were represented by more than block, so the model was not completely accurate. To remedy this, we have now clarified the nested nature of the data in the methods section and we have also made it explicit in the statistical model used.

Experimental block is nested within Accession, which again is nested within Population, resulting in the following model:

Survival ~ Population + (1|Population:Accession) + (1|Population:Accession:Block)

We have recalculated the p-values using this model and updated the results and supplemental tables accordingly. The updated p-values are higher in some cases but remain significant, so the conclusions are not affected. We have emphasized that the conclusions are drawn based on a limited set of accessions, by stating that: “Although it includes relatively few accessions, our reciprocal garden experiment clearly indicates that pop3 accessions are not generally more hardy in transplantation experiments and provides evidence to support that pop3 could be locally adapted to a cold climate.”

Yes, we have also calculated the Population x Location interaction p-value, which is highly significant ($p=3e-9$). We have now described this in the Results and Methods sections.

Some additional comments on presentation:

2) Line 103. Not clear why a weak link between geographic and genetic distance would complicate identification of local adaptation. On the contrary it could be argued that a correlation between geographic distance and measures of differences in environmental conditions complicates the detection of signals of selection (cf. necessary control of population structure).

Response to 2)

This point was discussed in a previous response, where we argued that “This is because human dispersal uncouples the plant genotype from its true geographical origin, which complicates interpretation of analyses of local adaptation.” We realize now that this clarification was not carried over into the manuscript text. We have addressed this by modifying the statement so it now reads “However, human dispersal of Arabidopsis seeds has weakened the link between geographic origin and plant genotype, complicating interpretation with respect to local adaptation”.

3) Line 257, “but worse...and the converse was true for non-pop3 individuals”. Change to “while the converse was true at the southern Miyazaki site”. Because your focus is on detecting evidence of local adaptation, the critical comparison is that between local and nonlocal

genotypes at each of the two sites (“pop3 vs. non-pop3 survival at each of the sites”). Revise the following sentence reporting pairwise tests accordingly.

Lines 260-263. Similarly, the description of performance of the different accessions at the two sites should focus on how local and nonlocal genotypes performed at each of the two sites (not on how nonlocal genotypes performed at the two sites).

Response to 3)

Yes, that is a good point. We have edited as suggested, which has clarified the focus and also made the paragraph easier to read. We have also added the result of the test for population x location interaction to this paragraph, see 1).

4) Line 267, “to maximize the chances of identifying phenotypic variation of adaptive relevance...”. Not clear what the documentation of flowering time in the greenhouse adds in this regard. Please, explain or remove.

Response to 4)

We have deleted the statement.

5) Line 268, “both a field and a greenhouse flowering time experiment”. Indicate where the field experiment was conducted. The previous paragraph discusses field data obtained at two different sites.

Response to 5)

We have indicated that the flowering time experiment was carried out at Tohoku.

6) Line 312. This sentence is difficult to understand. Consider possible rewordings.

Response to 6)

We deleted the sentence. The following sentence contains the required information.

7) *I think several of the section titles push conclusions beyond what can be stated based on the results presented and that they therefore should be moderated:*

“Population 3 is locally adapted to a cold climate” – I suggest change to something like

“Population 3 is adapted to a cold climate” or “Population 3 has higher tolerance to cold”

“Selection acted on winter survival during pop3 local adaptation” – Data are consistent with this, but do not prove it. I would suggest that a more appropriate title would be “Winter survival and pop3 differentiation”

“Flowering and seed traits influenced pop3 local adaptation” – Change to something like “pop3 has diverged in flowering phenology and seed traits”

Response to 7)

We have made the suggested changes.

8) *Line 382, “in a very regular fashion”. Unclear what this means.*

Response to 8)

We have deleted the text since it refers to “little long-distance gene flow post colonization” in the same sentence.

9) *Line 387. Insert “evidence of” before “local adaptation”.*

Response to 9)

The suggested change was made.

10) *Line 391, “nor did they account...” This is known since long and has been the topic of lots of papers before.*

Response to 10)

Yes, that is right. We do not claim novelty here, but mention it to highlight the benefits of combining the two types of analysis.

11) Line 407, “*adapted and non-adapted accessions*”. Change to “*local and nonlocal accessions*”.

Response to 11)

The suggested change was made.

12) Line 438. Insert “*during plant development*” after “*set in earlier*”.

Response to 12)

The suggested change was made.

13) Line 438, “*reduced the period of time Lotus seedlings had to establish before winter*”. Change to “*reduced the time available for Lotus seedlings to establish before winter*”.

Response to 13)

The suggested change was made

14) Line 576, “*each genotype was grown in multiple blocks of three individuals*”. Please, reword to state explicitly that the three plants in a block represented three different genotypes (if that was the case).

Response to 14)

That was not the case. We have clarified the experimental setup and data structure by stating: “The planting scheme, where each accession was grown in multiple blocks of three individuals, resulted in a nested structure of the experimental data with Block nested within Accession nested within Population.” This data structure is now explicitly reflected in the statistical analysis, see 1).

REVIEWERS' COMMENTS:

Reviewer #1 (Remarks to the Author):

The authors have carefully addressed the comments received on the previous version, and at this stage I have no further comments.

REVIEWERS' COMMENTS:

Reviewer #1 (Remarks to the Author):

The authors have carefully addressed the comments received on the previous version, and at this stage I have no further comments.

Response:

We thank the reviewer for carefully checking the revisions.